# Emergence and evolution of an interaction between intrinsically disordered proteins

**Greta Hultqvist[1]\*, Emma Åberg[1], Carlo Camilloni[2,3,4], Gustav N Sundell[1], Eva Andersson[1], Jakob Dogan[1,5], Celestine N Chi[6], Michele Vendruscolo[2]\*, Per Jemth[1]\***

[1]Department of Medical Biochemistry and Microbiology, Uppsala University, Uppsala, Sweden; [2]Department of Chemistry, University of Cambridge, Cambridge, United Kingdom; [3]Department of Chemistry, Technische Universität München, München, Germany; [4]Institute for Advanced Study, Technische Universität München, München, Germany; [5]Department of Biochemistry and Biophysics, Stockholm University, Stockholm, Sweden; [6]Laboratory of Physical Chemistry, Eidgenössische Technische Hochschule Zürich, Zürich, Switzerland

**Abstract** Protein-protein interactions involving intrinsically disordered proteins are important for cellular function and common in all organisms. However, it is not clear how such interactions emerge and evolve on a molecular level. We performed phylogenetic reconstruction, resurrection and biophysical characterization of two interacting disordered protein domains, CID and NCBD. CID appeared after the divergence of protostomes and deuterostomes 450–600 million years ago, while NCBD was present in the protostome/deuterostome ancestor. The most ancient CID/NCBD formed a relatively weak complex ($K_d \sim 5$ μM). At the time of the first vertebrate-specific whole genome duplication, the affinity had increased ($K_d \sim 200$ nM) and was maintained in further speciation. Experiments together with molecular modeling using NMR chemical shifts suggest that new interactions involving intrinsically disordered proteins may evolve via a low-affinity complex which is optimized by modulating direct interactions as well as dynamics, while tolerating several potentially disruptive mutations.

**\*For correspondence:** greta. hultqvist@pubcare.uu.se (GH); mv245@cam.ac.uk (MV); Per. Jemth@imbim.uu.se (PJ)

**Competing interests:** The authors declare that no competing interests exist.

## Introduction

While the majority of proteins fold into well-defined structures to function, a substantial fraction of the proteome is made up by intrinsically disordered proteins (IDPs) (*Uversky and Dunker, 2012*). These IDPs, which can be fully disordered or contain disordered regions of variable size, play pivotal roles in biology, usually by participating in protein-protein interactions that govern key functions such as transcription and cell-cycle regulation (*Uversky et al., 2008*; *Wright and Dyson, 2015*). IDPs are present in all domains of life, but they are more common and have a unique profile in eukaryotes as compared to archea and bacteria (*Peng et al., 2015*). One particular problem with analyzing structure-function relationships in IDPs with regard to evolution is that IDPs appear to evolve faster than structured proteins and are more permissive to substitutions that do not apparently modulate function in either a positive or negative way (*Brown et al., 2011*). In addition, insertion and deletion of amino acids are more common in IDPs (*Brown et al., 2010*; *Light et al., 2013*), further complicating structure-function analysis. Thus, while analyses of protein sequences have shed light on the evolution of IDPs, few biophysical studies have directly addressed the evolution of IDPs on a molecular level.

**eLife digest** Proteins are an important building block of life and are vital for almost every process that keeps cells alive. These molecules are made from chains of smaller molecules called amino acids linked together. The specific order of amino acids in a protein determines its shape and structure, which in turn controls what the protein can do. However, a group of proteins called 'intrinsically disordered proteins' are flexible in their shape and lack a stable three-dimensional structure. Yet, these proteins play important roles in many processes that require the protein to interact with a number of other proteins.

At multiple time points during evolution, new or modified proteins – and consequently new potential interactions between proteins – have emerged. Often, an interaction that is specific for a group of organisms has evolved a long time ago and not changed since. As intrinsically disordered proteins lack a specific shape, it is harder to study how their structure (or lack of it) influences their purpose; until now, it was not known how their interactions emerge and evolve.

Hultqvist et al. analyzed the amino acid sequences of two specific intrinsically disordered proteins from different organisms to reconstruct the versions of the proteins that were likely found in their common ancestors 450-600 million years ago. The ancestral proteins were then 'resurrected' by recreating them in test tubes and their characteristics and properties analyzed with experimental and computational biophysical methods.

The results showed that the ancestral proteins created weaker bonds between them compared to more 'modern' ones, and were more flexible even when bound together. However, once their connection had evolved, the bonds became stronger and were maintained even when the organism diversified into new species.

The findings shed light on fundamental principles of how new protein-protein interactions emerge and evolve on a molecular level. This suggests that an originally weak and dynamic interaction is relatively quickly turned into a tighter one by random mutations and natural selection. A next step for the future will be to investigate how other protein-protein interactions have evolved and to identify general underlying patterns. A deeper knowledge of how this molecular evolution happened will broaden our understanding of present day protein-protein interactions and might aid the design of drugs that can mimic proteins.

Here, we use an 'evolutionary biochemistry' approach (*Bridgham et al., 2006*; *Harms and Thornton, 2013*; *McKeown et al., 2014*), in which phylogenetic reconstruction of ancestral sequences is combined with biophysical experiments (*Figure 1*), to reconstruct the evolution of a particular protein-protein interaction. This interaction involves disordered protein domains from two different transcriptional coactivators: (*i*) the molten-globule-like (*Kjaergaard et al., 2010a*) nuclear co-activator binding domain (NCBD) present in the two paralogs CREB-binding protein (CREBBP, also known as CBP) and p300, and (*ii*) the highly disordered CREBBP-interacting domain (CID), found in the three mammalian paralogs NCOA1, 2 and 3 (also called SRC1, TIF2 and ACTR, respectively). The interaction between CID from NCOA3 (ACTR) and NCBD from CREBBP represents a classical example of coupled folding and binding of disordered protein domains (*Demarest et al., 2002*).

The two paralogs CREBBP and p300, as well as the three paralogs NCOA1, 2 and 3, likely result from whole genome duplications. In general, a gene family is created by speciation events and genomic duplications. The general conclusion is that the ancestral vertebrate(s) went through two rounds of whole genome duplications (denoted 1R and 2R, respectively) prior to the origin of jawed vertebrates, and that teleost (bony) fish experienced a third whole genome duplication (3R) (*Hughes and Liberles, 2008*; *Van de Peer et al., 2009*). Thus, if no gene losses had occurred, vertebrates would have four copies and teleost fish eight copies of every gene. Genomic duplications are beneficial for inventing new biochemical functions (*Ohno, 1970*; *Näsvall et al., 2012*). Therefore, the abundance of new genes following whole genome duplications creates multiple possibilities to evolve new functions on a large scale through point mutations and natural selection, and the 1R and 2R genome duplications have likely contributed to the broad repertoire of IDPs in vertebrates.

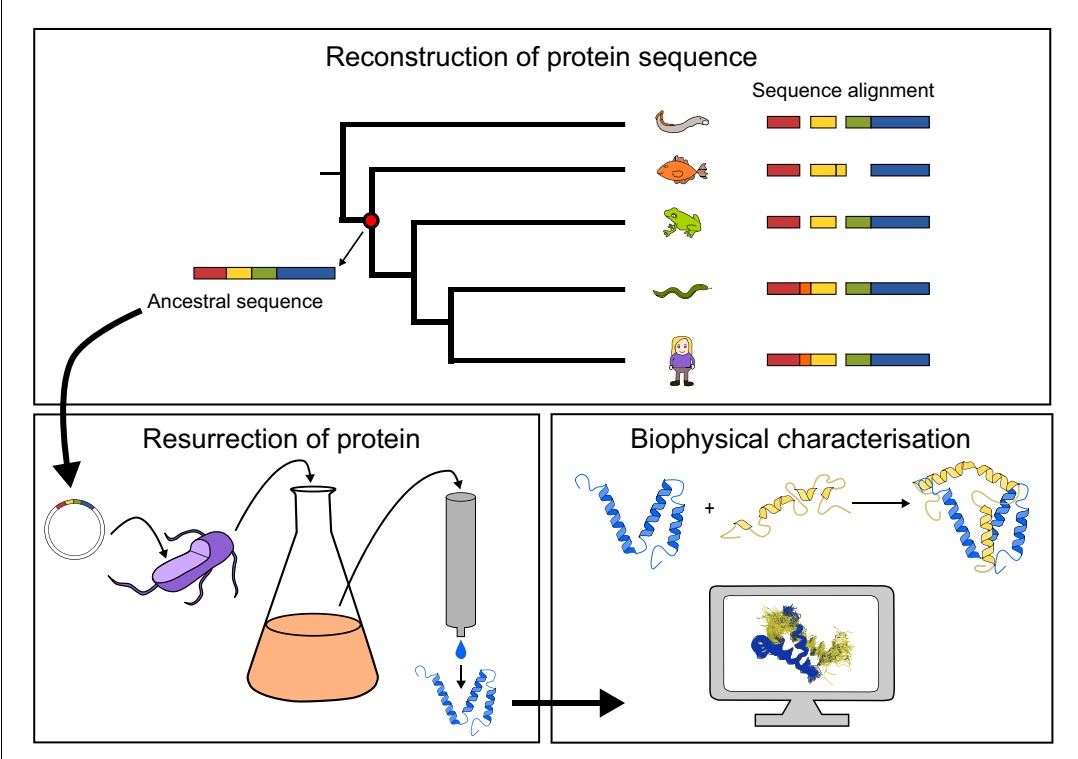

**Figure 1.** General approach to investigate the evolution of a protein-protein interaction involving intrinsically disordered domains. Multiple sequence alignment forms the basis for the phylogeny, which is used to predict ancient variants of two interacting protein domains, CID and NCBD, respectively. The ancient variants are then resurrected by expression in *Escherichia coli* and purified to homogeneity. Finally, the resurrected as well as present-day variants of CID and NCBD are subjected to biophysical and computational characterization to assess the evolution of structure-function relationships.

In the present paper, we track the evolutionary history during 600 million years (Myr) of two interacting IDPs, the older NCBD domain and the younger CID domain, with regard to sequence, affinity and structural properties. The two domains established a molecular interaction involved in regulation of transcriptional activation in the lineage leading to present day deuterostome animals, and may serve as a paradigmatic example of evolution of protein-protein interactions involving IDPs.

## Results

### Reconstruction of ancient sequences

As a first step in our study, ancestral versions of the amino acid sequences of the NCBD domain of CREBBP/p300 (NCBD) and the CID domain of NCOA (CID) were reconstructed by phylogenetic analyses (*Figure 2*, *Figure 2—figure supplements 1–4* and *Figure 2—source data 1* and *2*) using a maximum likelihood method (Materials and methods). This analysis shows that NCBD is older than CID, as NCBD can be traced in deuterostomes, protostomes (including extant arthropods, nematodes and molluscs) and cnidarians, but not in more distantly related eukaryotes such as the choanoflagellate *Monosiga brevicollis*. Hence, NCBD arose in the animal lineage as a domain within the ancestral CREBBP/p300 protein. CID, on the other hand, appeared as an IDP domain within NCOA after the split of deuterostomes and protostomes, since it could not be identified in protostome NCOA. Furthermore, CID was present in an early deuterostome, before the first whole-genome duplication (1R), since it could be traced in extant sea urchins and acorn worms, in addition to vertebrates. Thus, we conclude that the CID/NCBD interaction emerged at the beginning of the deuterostome lineage around the Cambrian period, an era with a very dramatic evolutionary history.

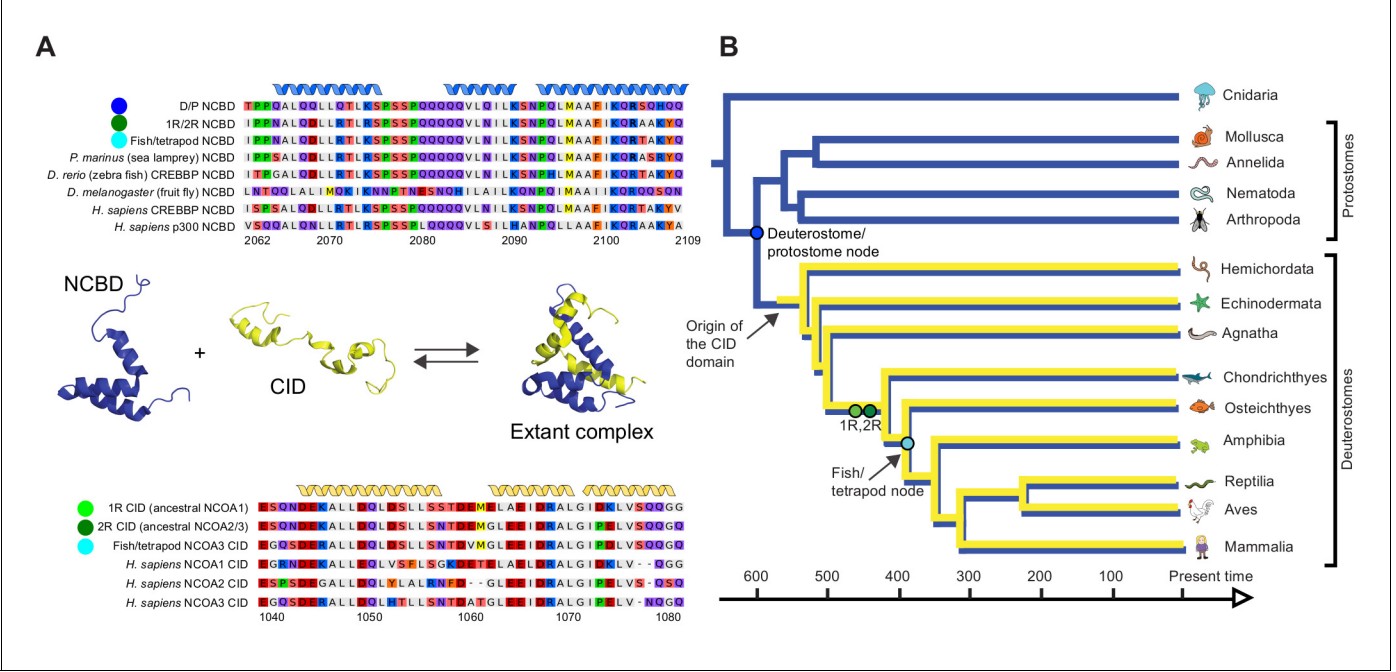

**Figure 2.** Reconstruction of the evolution of the interacting NCBD and CID domains. (A) Sequence alignments of extant and reconstructed ancient NCBD (top) and CID domains (bottom). The positions of helices are according to the NMR structure of the complex between extant CREBBP NCBD (blue) and NCOA3 CID (yellow). Free NCBD (protein data base code 2KKJ) and the CID/NCBD complex (1KBH) are NMR structures, whereas the picture of free CID is a hypothetical modified structure made from the NCOA1 CID/NCBD complex (2C52). The first residue in the NCBD alignment is referred to as position 2062 in the text and the first residue in the CID alignment as 1040. The color coding of the sequences reflects similarities in chemical properties of the amino acid side chains and is a guide for the eye to see patterns of conservation. (B) Schematic tree of life with selected animal groups depicting the evolution of the NCBD domain (blue) in both protostomes and deuterostomes and the CID domain (yellow) in the deuterostome lineage only. See *Figure 2—figure supplements 1–4* for detailed alignments and trees.

The following source data and figure supplements are available for figure 2:

**Source data 1.** Probabilities of resurrected amino acid residues at the respective position (2062–2109) in the NCBD domain.

**Source data 2.** Probabilities of resurrected amino acid residues at the respective position (1040–1081) in the CID domain.

**Figure supplement 1.** Sequence alignment of NCBD domains of CREBBP/p300 used in the phylogenetic reconstruction.

**Figure supplement 2.** Sequence alignment of the CID domains of NCOA1-3 used in the phylogenetic reconstruction.

**Figure supplement 3.** Phylogenetic tree of CREBBP/p300 proteins that contain the NCBD domain.

**Figure supplement 4.** Phylogenetic tree of NCOA1-3 proteins that contain the CID domain.

For the oldest reconstructed version of NCBD, deuterostome/protostome (D/P) NCBD, there were several relatively uncertain positions with regard to amino acid identity (*Figure 2—source data 1*). This was experimentally resolved by testing the effect of alternative residues at these positions in the context of one of the most likely D/P variants. The first position included in the NCBD domain (2062) was particularly problematic in this respect, with several amino acid identities with similar and low probability: Ile (0.19), Val (0.16), Thr (0.15), Met (0.13), etc. In addition, the probability numbers at this position were very sensitive to which extant sequences were included in the phylogenetic analysis. Because of this and to avoid a hydrophobic side-chain at the N-terminus, we chose Thr at position 2062 as the 'wild type' ancestral NCBD (referred to as D/P NCBD in the paper).

## Do the intrinsically disordered domains in NCOA and CREBBP have a higher amino acid substitution rate than the ordered domains?

Previous studies have suggested that the amino acid sequence of IDPs display a higher substitution rate (following selection) than those of ordered proteins (*Brown et al., 2011*). Although limited to two proteins, NCOA and CREBBP/p300, our data set allowed comparison between ordered and disordered regions within the same protein and during time. We therefore assessed the number of substitutions in selected ordered and disordered domains, respectively, in both NCOA and CREBBP/ p300. Clearly, amino acid sequences within linker regions between interaction domains have evolved such that no sequence similarity can be detected in most cases. For example, outside of the interacting regions of CID and NCBD, as defined by the NMR structures, it is impossible to align the sequences, even for closely related species. Thus, the domain boundaries were defined based on sequence conservation and available crystal or NMR structures. Based on how well we could define these boundaries and the confidence in the resurrection, which in some cases was low due to poor alignment quality, we selected four folded CREBBP domains (HAT, KIX, RING/PHD, TAZ1) and one folded NCOA domain (Pas-A) for the analysis. To make the comparison between folded and disordered protein interaction domains simple, we used the same alignments and phylogenetic trees used to reconstruct ancient CID and NCBD domains and reconstructed other domains from NCOA and CREBBP/p300, respectively. We then counted the number of amino acid substitutions and insertions/deletions in a particular sequence as compared to its predecessor. Thus, for domains from the CREBBP lineage, we compared the 1R/2R sequence with the D/P sequence, the ancestral fish/tetrapod (F/T) CREBBP with the 1R/2R, and present day human and zebrafish CREBBP with F/T CREBBP (*Figure 3*). In general, the amino acid substitution rates in NCOA and CREBBP/p300 are higher for all domains during the early times of animal evolution and up to the common ancestor of fish and tetrapods 390 Myr ago. Within CREBBP, the substitution rate of NCBD is similar to those of the ordered histone acetyltransferase (HAT) and RING/PHD domains, whereas KIX and TAZ1 have remained more conserved. For NCOA3 it was more difficult to define domain boundaries and we only compared CID to one folded domain, Pas-A. The overall substitution rate is only slightly higher for the CID domain, and the two domains have similar profiles. Thus, for CID and NCBD, functional constraints of the domains rather than disorder per se is probably determining amino acid

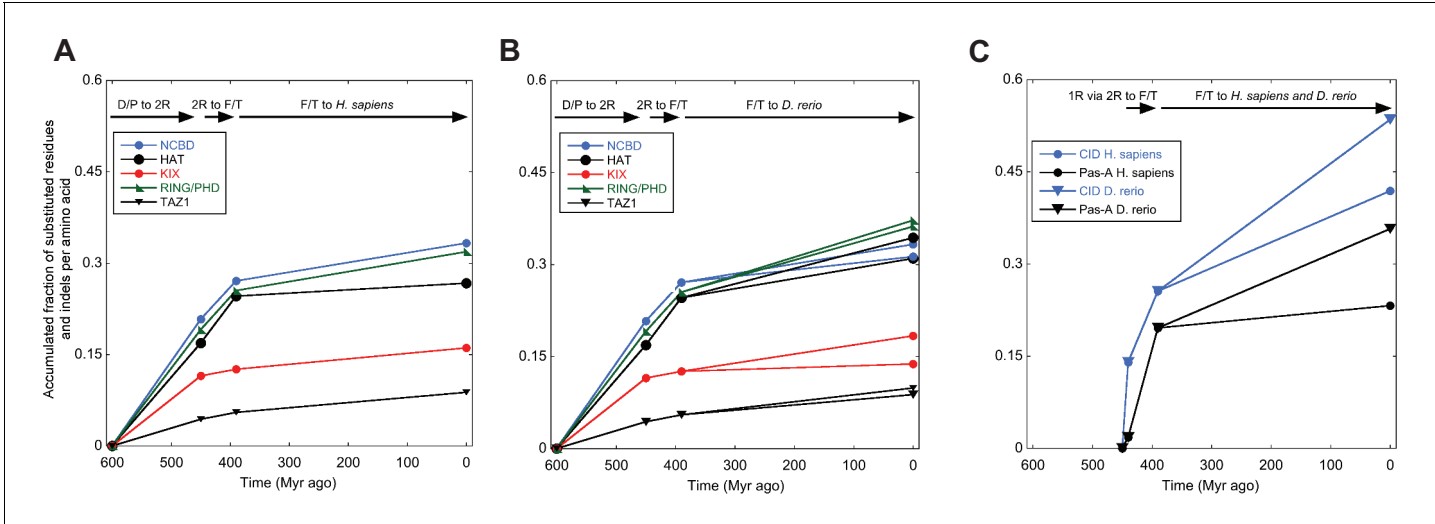

**Figure 3.** Amino acid substitutions in different domains in CREBBP/p300 and NCOA as a function of time. The predicted ancient sequences for distinct domains in CREBBP/p300 (**A** and **B**) and NCOA (**C**) were used to calculate the number of substitutions and indels between each evolutionary node (Deuterostome/protostome, D/P; 1R; 2R; Fish/tetrapod, F/T; and present day) in a particular lineage (human and zebrafish CREBBP and human and zebrafish NCOA3, respectively). The alignment and trees used to resurrect HAT, KIX, RING/PHD and TAZ1 were the ones optimized for NCBD. Similarly, the alignment and trees used to resurrect Pas-A were the ones optimized for the CID domain. The number of substitutions plus indels were normalized against the number of amino acid residues in each domain and the accumulated fraction of sequence changes plotted versus historical time. Both 1R and 2R occurred around 450 million Myr ago and the distance between them in panel **C** (10 Myr) is arbitrary.

substitution rate, similarly to a subset of disordered regions highlighted in earlier studies (*Brown et al., 2011*, *2010*, *2002*).

## Structural context of the CID and NCBD sequences

To bring the analysis from the sequence level to the structural level, we considered the NMR structures of extant mouse CREBBP NCBD in complex with human NCOA1 CID (*Waters et al., 2006*) and NCOA3 CID (*Demarest et al., 2002*), respectively. A structural alignment shows that the NCBD domain aligns well in the two complexes but that only the first α-helix (Cα1) of NCOA1 CID and NCOA3 CID occupy similar positions (*Figure 4*). In fact, the third α-helix (Cα3) of NCOA1 CID aligns with the second α-helix (Cα2) of NCOA3 CID, while Cα2 of NCOA1 CID is non-binding and split into two smaller α-helices. The first α-helix of NCBD (Nα1) interacts with Cα1 in both complexes, but the conformation of the third α-helix (Nα3) is slightly different in the two complexes. Residues forming the Nα1/Cα1 interface in the CID/NCBD complex are conserved in the deuterostome lineage, while Nα3 has accumulated five substitutions (*Figure 2A*).

## Determination of affinity between ancient and extant CID/NCBD variants

The next step in our approach was to 'resurrect' the sequences identified through our evolutionary analysis to analyze their biophysical properties. In these experimental studies, ancient and selected extant variants of CID and NCBD were expressed in *E. coli*, purified to homogeneity (see Materials and methods) and subjected to binding studies using isothermal titration calorimetry (ITC) to monitor changes in affinity over historical time. Strikingly, the affinity of deuterostome/protostome (D/P) NCBD for 1R CID, which is the oldest CID domain that we were able to resurrect with good confidence, was relatively low (5 μM) as compared to younger NCBD variants (*Figure 5A and B*, *Table 1*). Importantly, all tested ancestral D/P NCBD (Thr2062) variants measured with 1R CID yielded a relatively low affinity ($K_d$ values = 1.5 to 18 μM). The $K_d$ values for D/P NCBD with Ile and Val at position 2062 were 2.0 and 2.2 μM, respectively, which is very close to that of 'wild-type' D/P NCBD with Thr2062 (3.0 μM) (*Figure 5A* and *Figure 6*). Thus, our conclusions in the paper hold irrespective of the nature of the amino acid residues at the uncertain positions in D/P NCBD.

Already around the time of the two vertebrate-specific whole genome duplications (1R and 2R, respectively), NCBD had evolved an affinity toward the CID domain similar to that found today

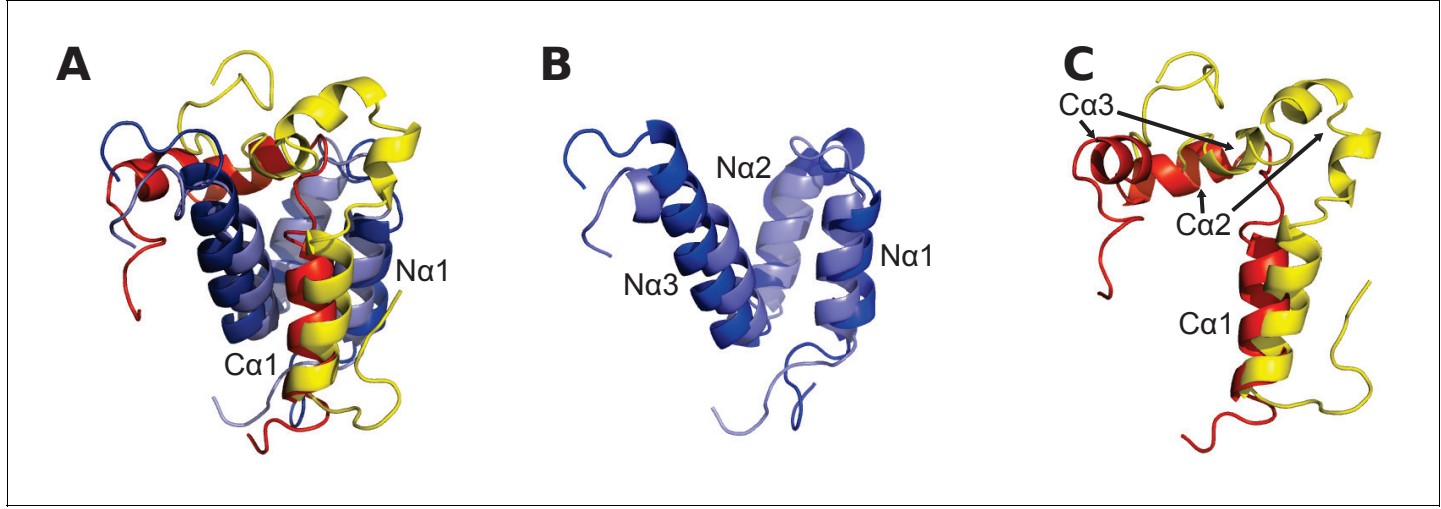

**Figure 4.** Structural alignment of two CID/NCBD complexes. (**A**) Superimposition of the structures of two complexes solved by NMR: CREBBP NCBD (Light blue)-NCOA1 CID (Yellow) (2C52) and CREBBP NCBD (Dark blue)-NCOA3 CID (Red) (1KBH). The complexes contain a hydrophobic core formed by residues from the respective protein domain. (**B**) Superimposition of NCBD from the complexes shows that in particular Nα1 and Nα2 align very well. (**C**) Superimposition of the NCBD-bound conformations of NCOA1 CID and NCOA3 CID. Whereas Cα1 from both complexes align well, the C-terminal regions of the CID domains occupy different positions.

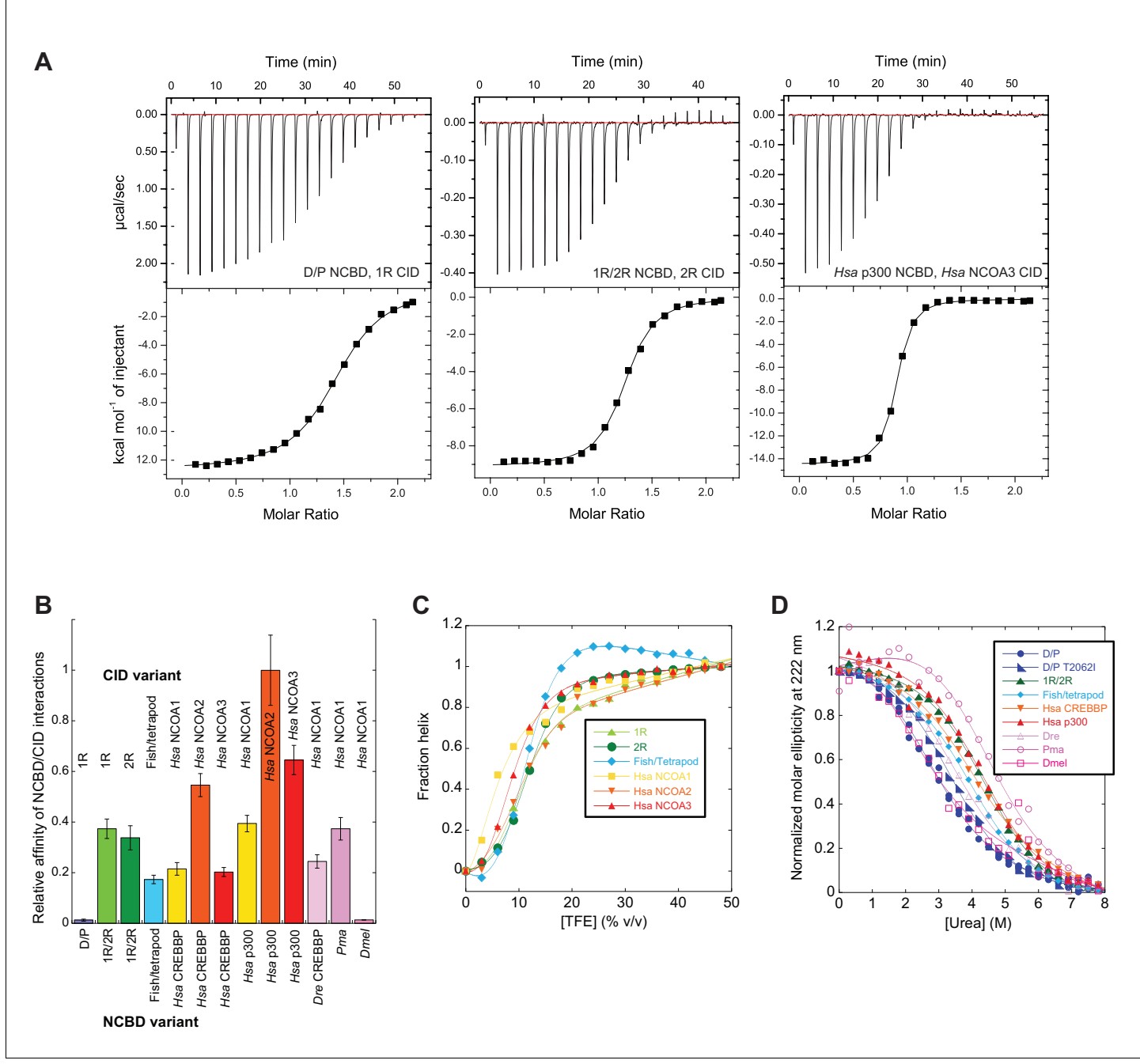

**Figure 5.** Biophysical characterization of ancient and extant CID and NCBD domains. (**A**) Affinity of CID/NCBD complexes was measured by isothermal titration calorimetry (three examples are shown including the low-affinity D/P NCBD, 1R CID interaction). (**B**) The affinities ($K_d$ values) were normalized against the interaction between extant human NCOA2 CID and p300 NCBD. The relative affinity for D/P NCBD, 1R CID was calculated using the average $K_d$ values of all D/P NCBD variants ($5 \pm 2$ μM). (**C**) Propensity for helix formation for ancient and extant CID domains as measured by circular dichroism at 222 nm upon addition of the helix stabilizer 1,1,1-trifluoroethanol. (**D**) Global stability of NCBD domains as measured by circular dichroism at 222 nm (reflecting the fraction folded NCBD) upon addition of the denaturant urea. *Hsa*, *Homo sapiens*; *Dre*, *Danio rerio* (zebrafish); *Pme*, *Petromyzon marinus*, (sea lamprey); *Dmel*, *Drosophila melanogaster* (fruit fly).

**Table 1.** Equilibrium dissociation ($K_d$ ±standard error) values for the interaction between NCBD and CID variants as determined by ITC.

| | Hsa NCOA1 CID (SRC1) | Hsa NCOA2 CID (TIF2) | Hsa NCOA3 CID (ACTR) | Fish/Tetrapod NCOA3 CID | 2R CID | 2R CID N1043S | 2R CID G1080S | 1R CID | 1R CID S1058N | 1R CID G1080S | 1R CID S1078Q | Hsa p53TAD | Hsa ETS-2 PNT |
|---|---|---|---|---|---|---|---|---|---|---|---|---|---|
| | $K_d$ ( µM) | | | | | | | | | | | | |
| Hsa CREBBP NCBD | 0.33 ± 0.039 | 0.13 ± 0.011 | 0.35 ± 0.031 | | 0.65 ± 0.024 | | | 0.38 ± 0.020 | | | | 84 ± 2.3 | 0.76 ± 0.071 |
| Hsa p300 NCBD | 0.18 ± 0.015 | 0.071 ± 0.010 | 0.11 ± 0.010 | | 0.28 ± 0.012 | | | 0.22 ± 0.024 | | | | 9.2 ± 2.2 | 1.5 ± 0.077 |
| Dre CREBBP NCBD | 0.29 ± 0.032 | 0.23 ± 0.013 | 0.63 ± 0.057 | 0.57 ± 0.025 | | | | | | | | | |
| Pma NCBD | 0.19 ± 0.023 | 0.044 ± 0.017 | 0.23 ± 0.030 | | | | | | | | | | 1.0 ± 0.10 |
| Dmel NCBD | 5.2 ± 0.20 | 22 ± 1.6 | 37 ± 2.8 | | 4.1 ± 0.93 | | | 9.7 ± 1.6 | | | | | No detectable binding |
| Fish/Tetrapod CREBBP NCBD | | | | 0.41 ± 0.040 | | | | | | | | 52 ± 5.2 | 1.3 ± 0.083 |
| 1R/2R NCBD | 0.11 ± 0.042 | 0.045 ± 0.018 | 0.23 ± 0.040 | | 0.28 ± 0.021 | 0.290 ± 0.035 | 0.33 ± 0.023 | 0.20 ± 0.016 | 0.22 ± 0.027 | 0.24 ± 0.024 | 0.25 ± 0.021 | 34 ± 4.0 nM | 0.85 ± 0.046 |
| 1R/2R NCBD N2065S | | | | | 0.11 ± 0.020 | 0.15 ± 0.013 | | 0.13 ± 0.012 | | | | | |
| 1R/2R NCBD N2065S K2107R | | | | | 0.18 ± 0.021 | 0.160 ± 0.011 | | 0.17 ± 0.023 | | 0.13 ± 0.018 | | | |
| D/P NCBD | 1.5 ± 0.088 | 0.52 ± 0.032 | 5.0 ± 0.22 | | | | | 3.0 ± 0.13 | 3.9 ± 0.16 | 4.8 ± 0.20 | 5.5 ± 0.21 | 43 ± 3.9 | 1.4 ± 0.051 |
| D/P NCBD T2062I | | | | | | | | 2.0 ± 0.2 | | | | | |
| D/P NCBD T2062V | | | | | | | | 2.2 ± 0.6 | | | | | |
| D/P NCBD P2063L | | | | | | | | 7.7 ± 0.53 | | | | | |
| D/P NCBD Q2088H | | | | | | | | 1.5 ± 0.080 | | | | | |
| D/P NCBD Q2088N | | | | | | | | 2.2 ± 0.070 | | | | | |
| D/P NCBD H2107Q | | | | | | | | 18 ± 1.2 | | | | | |
| Hsa CREBBP NCBD A2106Q | | 0.10 ± 0.02 | | | | | | | | | | | |
| Hsa CREBBP NCBD Y2108Q | | 0.21 ± 0.06 | | | | | | | | | | | |
| Hsa CREBBP NCBD A2106Q/Y2108Q | | 0.22 ± 0.06 | | | | | | | | | | | |

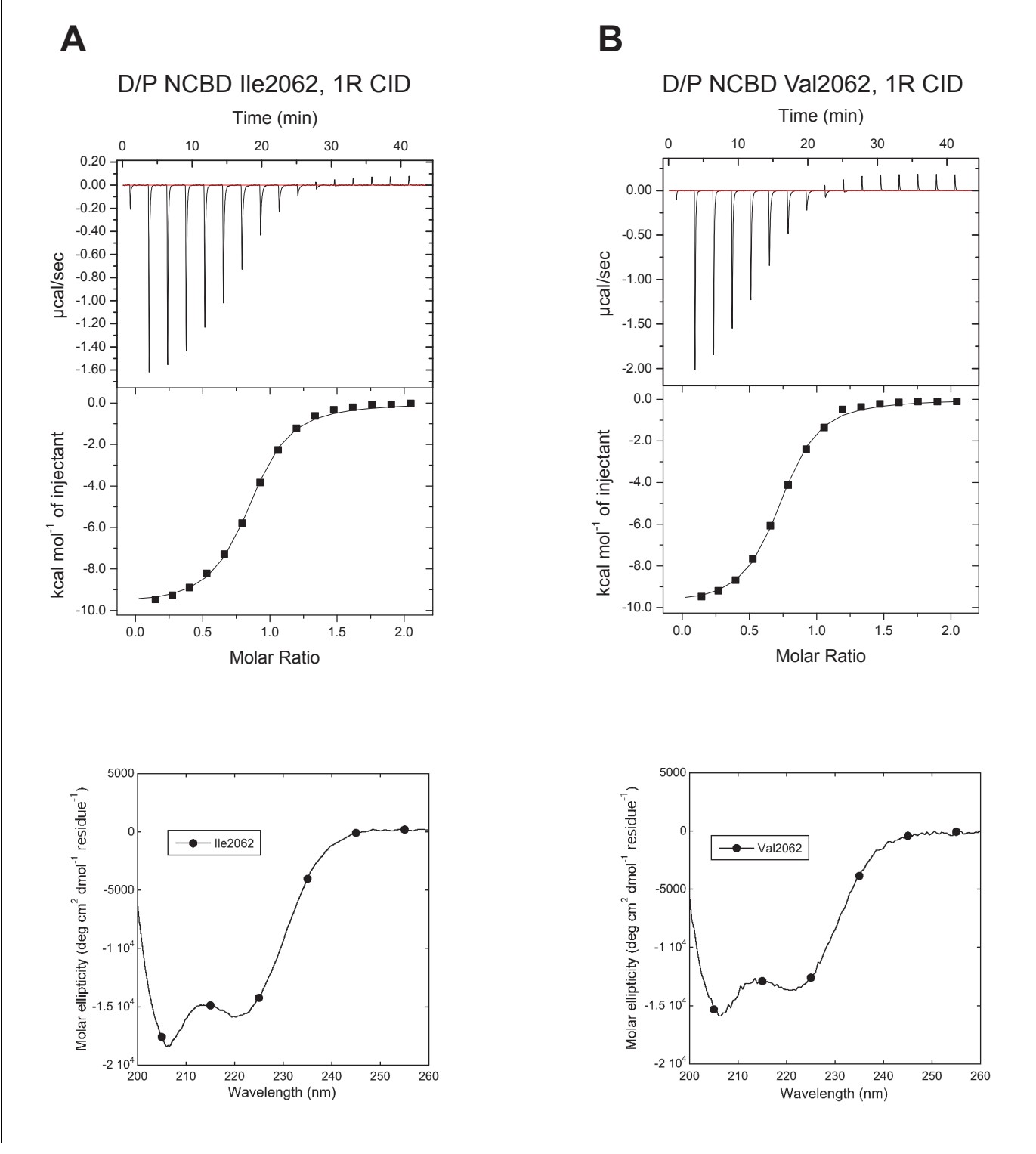

**Figure 6.** Characterization of alternative variants at position 2062 in D/P NCBD. Isothermal titration calorimeter and circular dichroism experiments of D/P NCBD with (**A**) Ile and (**B**) Val at position 2062. See *Figure 5A* for Thr2062 and *Table 1* for $K_d$ values.

(~200 nM). All tested alternative variants of 1R/2R NCBD, 1R CID and 2R CID yielded $K_d$ values in the same range (100–300 nM) (*Table 1*). Moreover, the affinities between extant NCBD domains and human CID domains were found to be surprisingly similar across several species. For example, CREBBP NCBD from sea lamprey and human, respectively, has similar affinity toward human NCOA3 CID. We also determined the affinity between ancient and extant NCBD domains and extant versions of other protein ligands for NCBD, namely the disordered transactivation domain (TAD) of human p53 (p53TAD) and the globular pointed (PNT) domain from human ETS-2. In contrast to the CID domain, the affinities of p53TAD and PNT, respectively, were similar for ancient and extant NCBD (*Table 1*).

## Structural stability of ancestral and extant variants of CID and NCBD

The NCOA3 CID domain from human displays a high degree of disorder (*Ebert et al., 2008*; *Kjaergaard et al., 2010b*) and all tested CID variants in the present study displayed a strikingly similar far-UV CD spectrum, typical of IDPs (*Figure 7A–C*). However, the propensity to form α-helices may differ between variants, which could result in higher affinity (*Iešmantavičius et al., 2014*). Addition of 1,1,1-trifluoroethanol (TFE) induces α-helix formation and can be used to experimentally assess the α-helical propensity of any given polypeptide when there is a helix-coil equilibrium (*Jasanoff and Fersht, 1994*). TFE titrations (0–50%) were performed for ancient and extant CID domains (*Figure 5C* and *Table 2*). The similar shape and midpoints of the resulting sigmoidal curves (reflecting the helix-coil transition) demonstrates that the overall α-helical propensity for ancient and modern CID variants is virtually identical. Increased α-helical propensity would likely have increased the affinity for NCBD (*Iešmantavičius et al., 2014*), but at the cost of lower plasticity and flexibility. We note, however, that the stability for individual α-helices as predicted by the software AGADIR (*Muñoz and Serrano, 1994*) has changed during evolution. For example, α-helix 1 of CID from human NCOA1 and 2 displays a higher α-helix propensity than α-helix 1 from human NCOA3 and the ancestral versions (*Figure 8*). Such local changes in α-helical propensity is a likely route to evolve an optimal affinity for short recognition motifs, which are common in intrinsically disordered regions (*Fuxreiter et al., 2004*).

While extant mammalian CREBBP NCBD has a small but well-defined hydrophobic core, it is a very dynamic protein domain in the absence of a bound CID domain as previously shown by NMR measurements (*Ebert et al., 2008*). We were therefore particularly interested in whether the global stability of NCBD changed when CID was recruited as binding partner. CD spectra (*Figure 7D–F*) and thermal denaturations (*Figure 7G–I*) were similar for all NCBD variants. The most ancient D/P NCBD variant was slightly less stable than historically subsequent variants in the deuterostome lineage, as determined by the midpoint from far UV CD-monitored urea denaturation experiments (*Figure 5D* and *Table 3*). However, despite high precision in experimental data, the broad unfolding transition of small protein domains such as NCBD leads to low accuracy in the estimates of the free energy of denaturation. Therefore, we refrain from making strong conclusions regarding the stability despite a clear shift in the urea midpoint for unfolding.

## NMR and molecular modeling of ancient and extant CID/NCBD complexes

We next asked what happens on a molecular level when a new binding partner is recruited. To shed light on this question, we first subjected two ancient complexes (1R CID with D/P NCBD and 1R CID with 1R/2R NCBD, respectively) and one extant complex (human NCOA3 CID with human CREBBP NCBD) to NMR experiments and then used the chemical shifts assigned for Cα, Cβ, H and N as restraints in molecular dynamics simulations using Metadynamic Metainference (*Bonomi et al., 2016a*, *2016b*), a recently developed scheme that can optimally balance the information contents of experimental data with that of a priori physico-chemical information.

Structural ensembles based on molecular dynamics simulations and NMR chemical shifts were obtained starting from the two available NMR structures, mouse CREBBP NCBD in complex with human NCOA1 CID (2C52) (*Waters et al., 2006*) and NCOA3 CID (1KBH) (*Demarest et al., 2002*), respectively. By using Metadynamic Metainference simulations (see Materials and methods), three complexes were analyzed: (*i*) 1R CID with, D/P NCBD, (*ii*) 1R CID with 1R/2R NCBD and (*iii*) extant human NCOA3 CID with CREBBP NCBD (*Figure 9*). The NMR data used as restraints in the

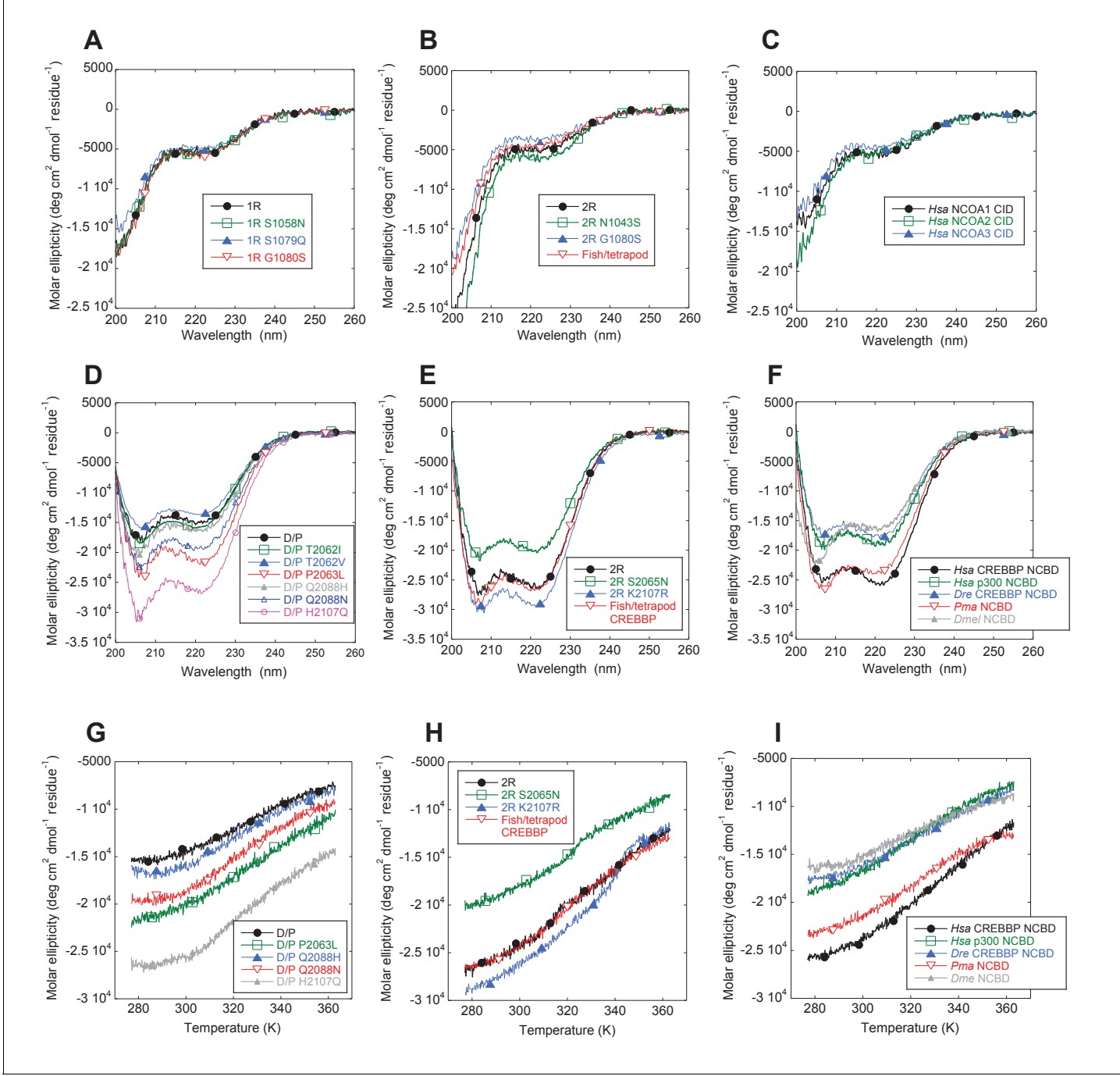

**Figure 7.** Far-UV Circular dichroism experiments. (A–C) CD spectra of CID variants display a profile typical for disordered proteins. (D–E) CD spectra of NCBD variants show a qualitatively similar shape for all variants. (G–I) Thermal denaturations of NCBD variants show a similar apparent non-cooperative transition.

simulations were of high quality and the peak assignment was close to 100% for all the three complexes (*Figure 9—figure supplement 1* and *Figure 9—source data 1*). Comparison of the structural ensembles for the three complexes and their relative free-energy projections indicates that evolution of higher affinity correlates with a somewhat reduced conformational heterogeneity (*Figure 9A*). Indeed, the free-energy surfaces as a function of the fraction of the overall helix-content and the radius of gyration (Rg) show that the ancient complex is slightly less structured and more compact,

**Table 2.** Equilibrium parameters for CD-monitored trifluoroethanol (TFE) induced helix formation of CID variants determined in 20 mM sodium phosphate, pH 7.4, 150 mM NaCl, at 25°C.

| CID variant | $[TFE]_{50\%}$* (%) | $[TFE]_{50\%}$† (%) | $m_{D-N}$† ($\%^{-1}$) |
|---|---|---|---|
| 1R‡ | 8.5 ± 1.3 | 7.6 ± 2.3 | 0.15 ± 0.02 |
| 2R | 10.7 ± 0.9 | 12.0 ± 0.2 | 0.22 ± 0.01 |
| Fish/tetrapod NCOA3 | 9.9 ± 0.7 | 10.0 ± 0.6 | 0.17 ± 0.01 |
| *Hsa* NCOA1 | _§ | _§ | 0.15 ± 0.03 |
| *Hsa* NCOA2 | 9.5 ± 1.7 | _§ | 0.11 ± 0.02 |
| *Hsa* NCOA3 | 5.6 ± 1.5 | 6.5 ± 0.9 | 0.18 ± 0.01 |

*The $m_{D-N}$ value was shared among the datasets in the curve fitting; $m_{D-N}$ = 0.17 ± 0.01 $\%^{-1}$.

†Free fitting of both $[TFE]_{50\%}$ and $m_{D-N}$.

‡1R, the node around the time of the first whole genome duplication in the vertebrate lineage; 2R, the node around the time of the second whole genome duplication in the vertebrate lineage; Fish/tetrapod, the node where fish diverged from tetrapods; *Hsa*, *Homo sapiens*; *Dre*, *Danio rerio* (zebrafish); *Pme*, *Petromyzon marinus*, (sea lamprey); *Dmel*, *Drosophila melanogaster* (fruit fly).

§Not well determined in the curve fitting.

with an average helical fraction of 0.41. This is reflected in the ensemble with more disordered N- and C-terminal helices for CID and a more disordered C-terminal helix Nα3 for NCBD. The younger 1R/2R complex is a little more structured in particular with respect to the C-terminal helix of NCBD, with an average helical fraction of 0.44. The extant human complex, finally, is again slightly more

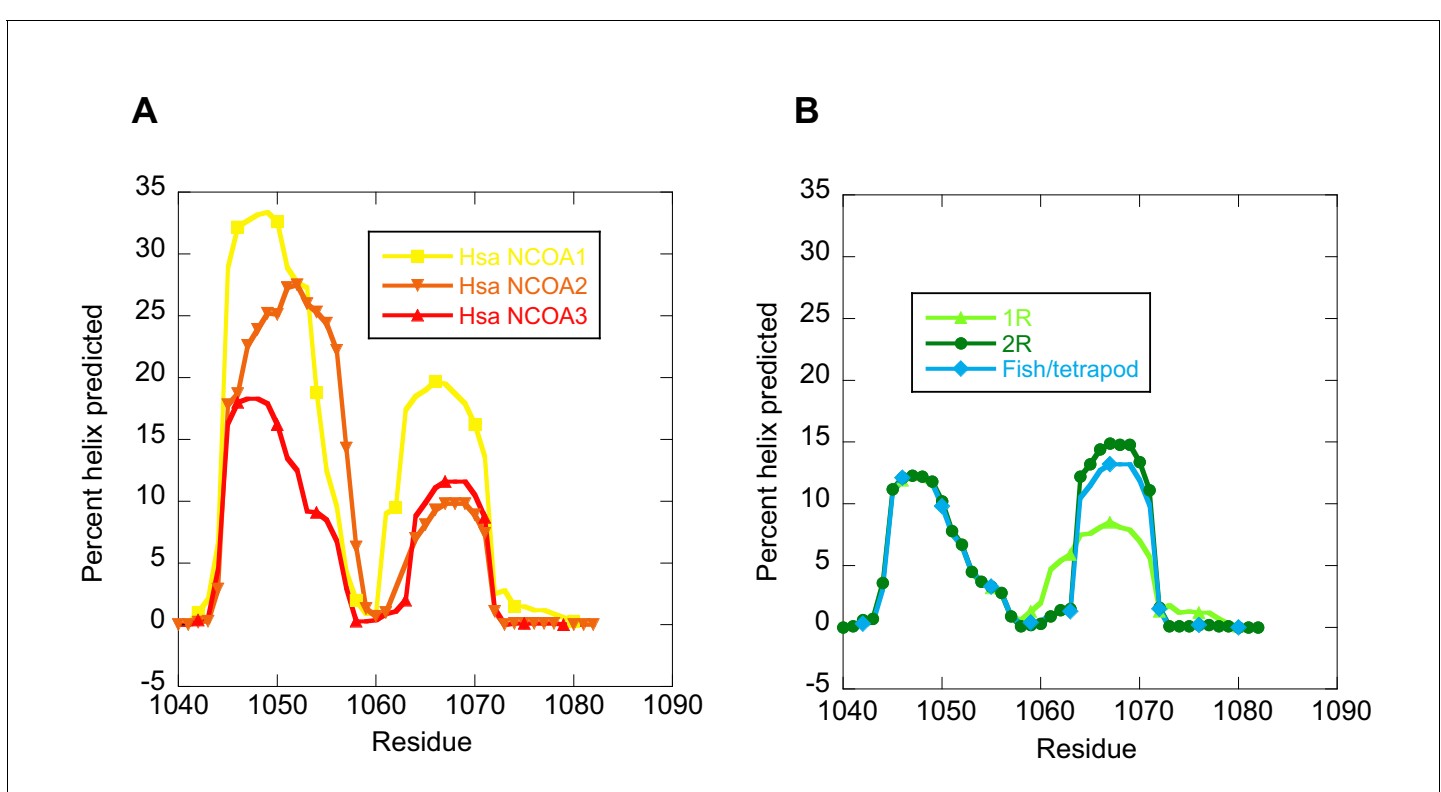

**Figure 8.** The helical propensity of CID variants as predicted by AGADIR. (**A**) CID domains from extant human NCOA1, 2 and 3. (**B**) Ancestral CID domains: 1R, 2R and the fish/tetrapod ancestor.

**Table 3.** Equilibrium parameters for CD-monitored urea denaturation of NCBD variants determined in 20 mM sodium phosphate, pH 7.4, 150 mM NaCl, 1 M TMAO at 10°C.

| NCBD variant | $[Urea]_{50\%}$[*] (M) | $\triangle G_{D-N}$[*] (kcal mol$^{-1}$) | $[Urea]_{50\%}$[†] (M) | $m_{D-N}$[†] (kcal mol$^{-1}$) | $\triangle G_{D-N}$[†] (kcal mol$^{-1}$) |
|---|---|---|---|---|---|
| D/P[‡] | 2.4 ± 0.4 | 1.5 ± 0.3 | 2.2 ± 0.2 | 0.56 ± 0.04 | 1.2 ± 0.2 |
| D/P T2062I | 3.3 ± 0.3 | 2.0 ± 0.3 | 3.4 ± 0.1 | 0.70 ± 0.08 | 2.4 ± 0.3 |
| 1R/2R | 4.4 ± 0.3 | 2.7 ± 0.3 | 4.4 ± 0.1 | 0.67 ± 0.05 | 3.0 ± 0.3 |
| Fish/tetrapod CREBBP | 4.0 ± 0.3 | 2.5 ± 0.3 | 4.0 ± 0.1 | 0.62 ± 0.05 | 2.5 ± 0.2 |
| *Hsa* CREBBP | 3.8 ± 0.3 | 2.3 ± 0.3 | 3.7 ± 0.2 | 0.46 ± 0.09 | 1.7 ± 0.4 |
| *Hsa* p300 | 4.4 ± 0.3 | 2.7 ± 0.3 | 4.4 ± 0.3 | 0.66 ± 0.17 | 2.9 ± 0.8 |
| *Dre* CREBBP1[§] | 3.4 ± 0.3 | 2.1 ± 0.3 | 2.2 ± 1.6 | 0.33 ± 0.16 | 0.7 ± 0.6 |
| *Pma* | 4.1 ± 0.2 | 2.5 ± 0.3 | 4.2 ± 0.6 | 0.50 ± 0.22 | 2.1 ± 1.0 |
| *Dmel* | 1.6 ± 0.5 | 1.0 ± 0.3 | 2.6 ± 0.4 | 1.2 ± 0.7 | 3.3 ± 1.9 |

[*]The $m_{D-N}$ value was shared among the datasets in the curve fitting; $m_{D-N} = 0.61 \pm 0.05$ kcal mol$^{-1}$M$^{-1}$.

[†]Free fitting of both $[Urea]_{50\%}$ and $m_{D-N}$

[‡]D/P, Deuterostome/protostome node; 1R/2R, the node(s) around the time of the two whole genome duplications in the vertebrate lineage; Fish/tetrapod, the node where fish diverged from tetrapods; *Hsa*, *Homo sapiens*; *Dre*, *Danio rerio* (zebrafish); *Pme*, *Petromyzon marinus*, (sea lamprey); *Dmel*, *Drosophila melanogaster* (fruit fly).

[§]The bony fish lineage experienced a third whole-genome duplication and has two variants of CREBBP NCBD.

structured at the N- and C- terminal helices of CID, with an average helical fraction of 0.47. These results were confirmed by an independent analysis of the chemical shifts by δ2D (*Camilloni et al., 2012a*) (*Figure 9B*) that shows how the terminal helices obtain more structure in going from the ancient (blue), via the 1R/2R complex (green) and to the extant human complex (red); yet, the helix content of the second helix of CID decreases. The increase in helical structure at the terminal helices corresponds to a decrease in the average fluctuations as was confirmed by the analysis of the root mean square fluctuations (RMSF) (*Figure 9C*). The changes in the helical structure have an effect in terms of inter-domain contacts. Indeed, the most ancient complex appears to have more but less populated contacts than younger complexes, in line with a marginally more disordered interface (*Figure 10*). During evolution, the total number of possible contacts between the domains decreases while the average population of the formed contacts increases. To visualize the extent to which different residues in NCBD interact in the respective CID complex (most ancient, 1R/2R, and extant CREBBP NCBD/human NCOA3 CID), the normalized number of interface contacts per residue was analyzed (*Figure 11*). In this analysis, we also included two other complexes involving extant CREBBP NCBD, those with p53TAD and the 'interferon regulatory factor (IRF) interaction domain from IRF-3' (denoted as IRF-3 in the paper), respectively. Overall, the main effect observed along the evolution of the CID/NCBD complex is a decreased fraction of contacts formed by N-terminal residues correlating with increased helical structure of the N-terminal helix of CID (*Figure 9*) and an increased fraction of contacts formed by C-terminal residues of NCBD, correlating with increased helical structure of this region (*Figure 9*). In the C-terminal, there is an increase in fraction of contacts in particular at position 2108 that joins Arg2104 in binding Asp1068 from CID, and at position 2105 whose side chain forms multiple hydrophobic contacts with CID. However, there is a decrease in fraction of contacts at position Gln2103 while other positions show a less clear trend reflecting the complexity of the changes in the interaction surface. Of note is the similarity of the interface used by NCBD to bind CID with that of the NCBD/p53TAD complex, where for example Arg2104 makes a salt-bridge with Asp49 while the Tyr2108 makes a hydrogen bond with the backbone of Met44, and its difference with the structurally distinct NCBD/IRF-3 complex, where the same two residues are exposed to the solvent.

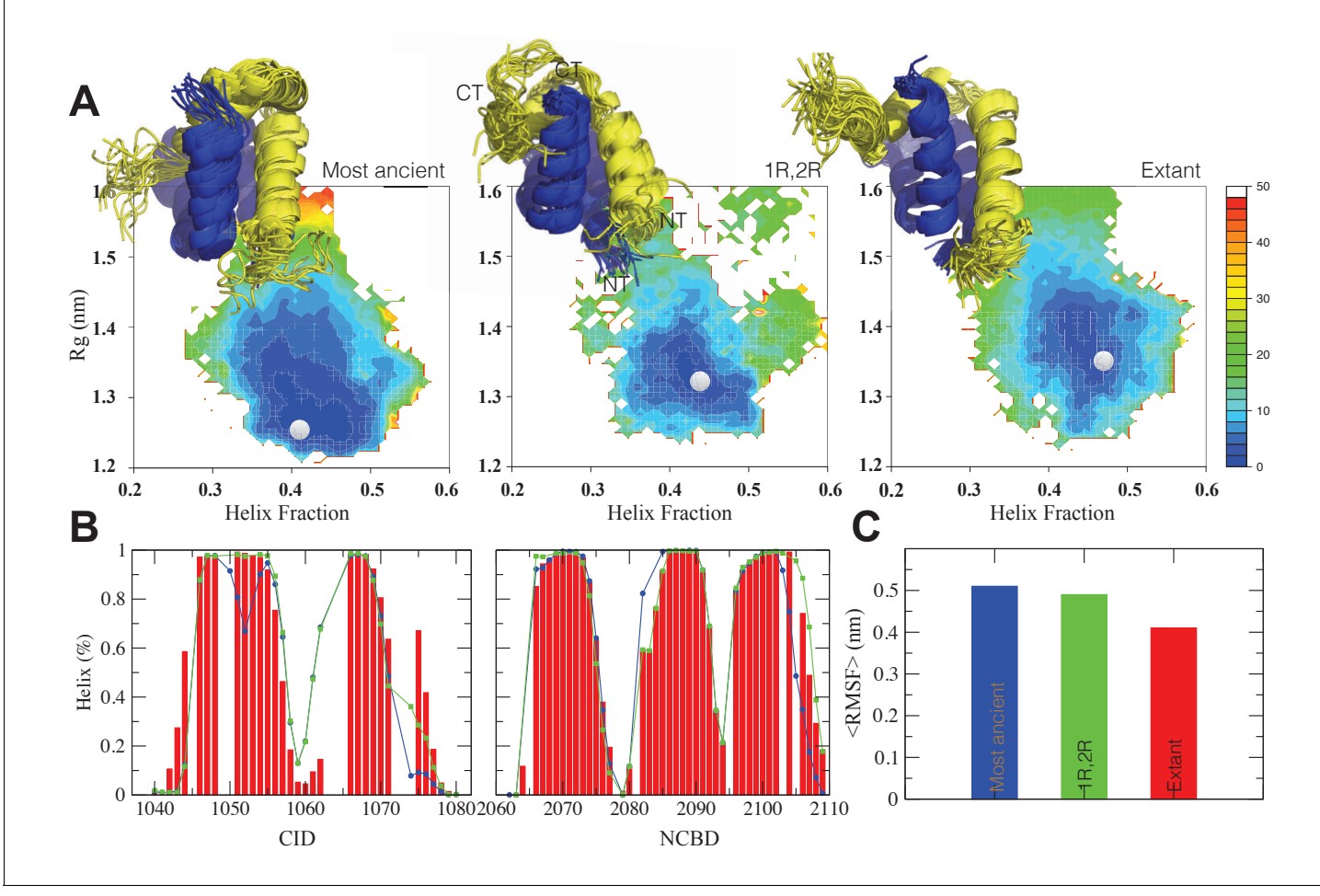

**Figure 9.** The CID/NCBD complex displays minor structural changes upon evolution. (**A**) Free-energy surfaces (in kJ/mol) as a function of the fraction of helix content and the Rg, for the most ancient complex (D/P NCBD and 1R CID), the 1R/2R complex and one extant complex (human NCOA3 CID/CREBBP NCBD). For each free-energy surface, the position of the minimum and a set of representative structures are shown: CID in yellow and NCBD in blue. N- and C- termini (NT and CT, respectively) are labeled for the central ensemble. (**B**) Per residue helix population of the protein ensembles of the most ancient (blue circles), 1R/2R (green squares) and extant (red bars) variants as predicted by δ2D from the chemical shifts. (**C**) Average root-mean-square fluctuation for the three variants showing a weak correlation between historical age and conformational heterogeneity of the complex.

The following source data and figure supplement are available for figure 9:

**Source data 1.** Chemical shift data of CID/NCBD complexes used in the molecular dynamics simulations
**Figure supplement 1.** Heteronuclear single quantum correlation ($^1$H/$^{15}$N) spectra for the ancient complex between 1R CID and D/P NCBD (red peaks) and the extant complex between human NCOA3 CID and CREBBP NCBD (blue peaks).

## The role of positions 2106 and 2108 in NCBD for increasing the affinity for the CID domain

The Metadynamic Metainference analysis highlighted a few positions that appeared prominent in the evolution of higher affinity in the CID/NCBD interaction and in particular position 2108, where the Q2108Y mutation makes the interaction stronger. To test this result and further probe the less conserved region at the end of Nα3 in NCBD, we made the two reverse mutations in human CREBBP NCBD (i.e. A2106Q, Y2108Q and the double mutant A2106Q/Y2108Q) and measured the affinity to human NCOA2 CID using ITC (*Figure 12*). The A2106Q mutation did not change the affinity for NCOA2 CID. The Y2108Q and the double mutation A2106Q/Y2108Q resulted in a slightly lower affinity (twofold) toward NCOA2 CID (*Table 1*), in support of the simulation. However, the

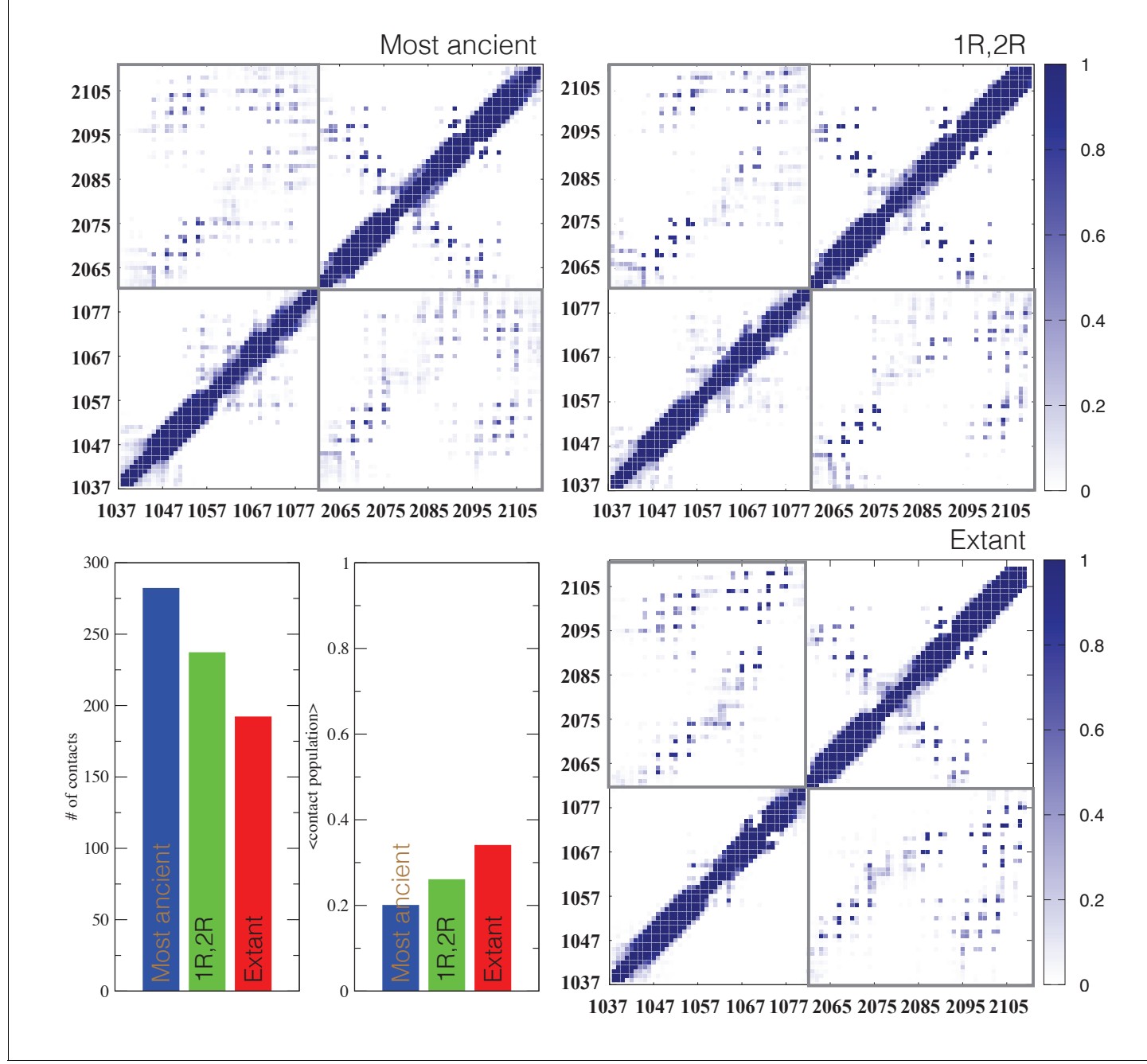

**Figure 10.** Contact analysis for ancient and extant CID/NCBD complexes. The probability contact maps are shown for each pair of residues for (upper left) the most ancient complex (1R CID and D/P NCBD), (upper right) the 1R/2R complex and (lower right) the extant NCOA3 CID/CREBBP NCBD complex. Inter-domain contacts are framed by gray rectangles. Given two residues in a certain conformation, a contact is defined as a distance within 0.5 nm (excluding hydrogen atoms). Lower left panels: The total number of inter-domain contacts (left) and the inter-domain average contact formation (right) are reported as the number of residues with a contact populated more than 5% and the average over population for the same contacts, respectively.

mutations made the NCBD protein less soluble, which led to precipitation and less reliable data due to ill-defined native baselines in the titration experiments. Nevertheless, the limited effect of these reverse mutations in one extant complex underscores the high degree of plasticity in this region of the CID/NCBD complex and illustrates the permissive nature of IDPs toward mutations.

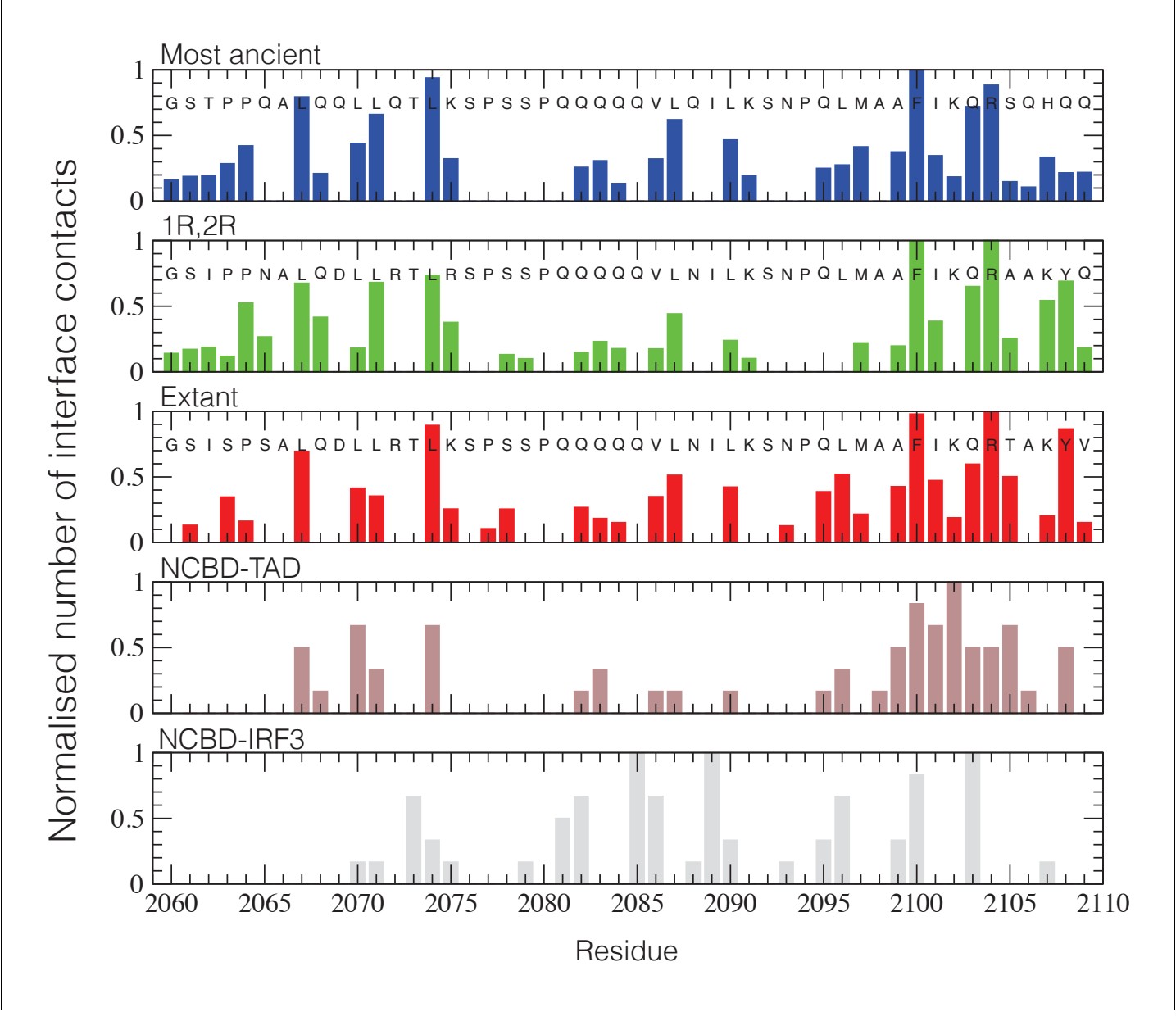

**Figure 11.** NCBD Interface contact analysis. The normalized number of interface contacts per residue is calculated from the simulations of the three historical CID/NCBD complexes (upper three panels) and compared with two extant complexes formed by CREBBP NCBD and alternative protein ligands, p53TAD (pdb code 2L14) (*Lee et al., 2010*) and a binding domain from IRF-3 (pdb code 1ZOQ) (*Qin et al., 2005*), respectively. In the IRF-3 complex (bottom panel), NCBD adopts a distinct tertiary structure as compared to complexes with CID and p53. The Gly-Ser residues at the N-terminus of the NCBD sequences result from the expression construct used in the study.

## Discussion

We have used evolutionary biochemistry to reconstruct the evolutionary process by which the interaction between two disordered proteins has emerged from a low-affinity complex. By identifying and resurrecting the early ancestor partners, we show how a combination of direct interactions and structural heterogeneity contributed to optimizing the affinity of the CID/NCBD complex. In accordance, we observe a remarkably high tolerance to mutation in the interface of this particular protein-protein interaction in extant complexes. For example, in NCBD there are a number of substituted residues at the end of Nα3 forming the interface to CID, (*Figure 2A*). For the CID domain, there are

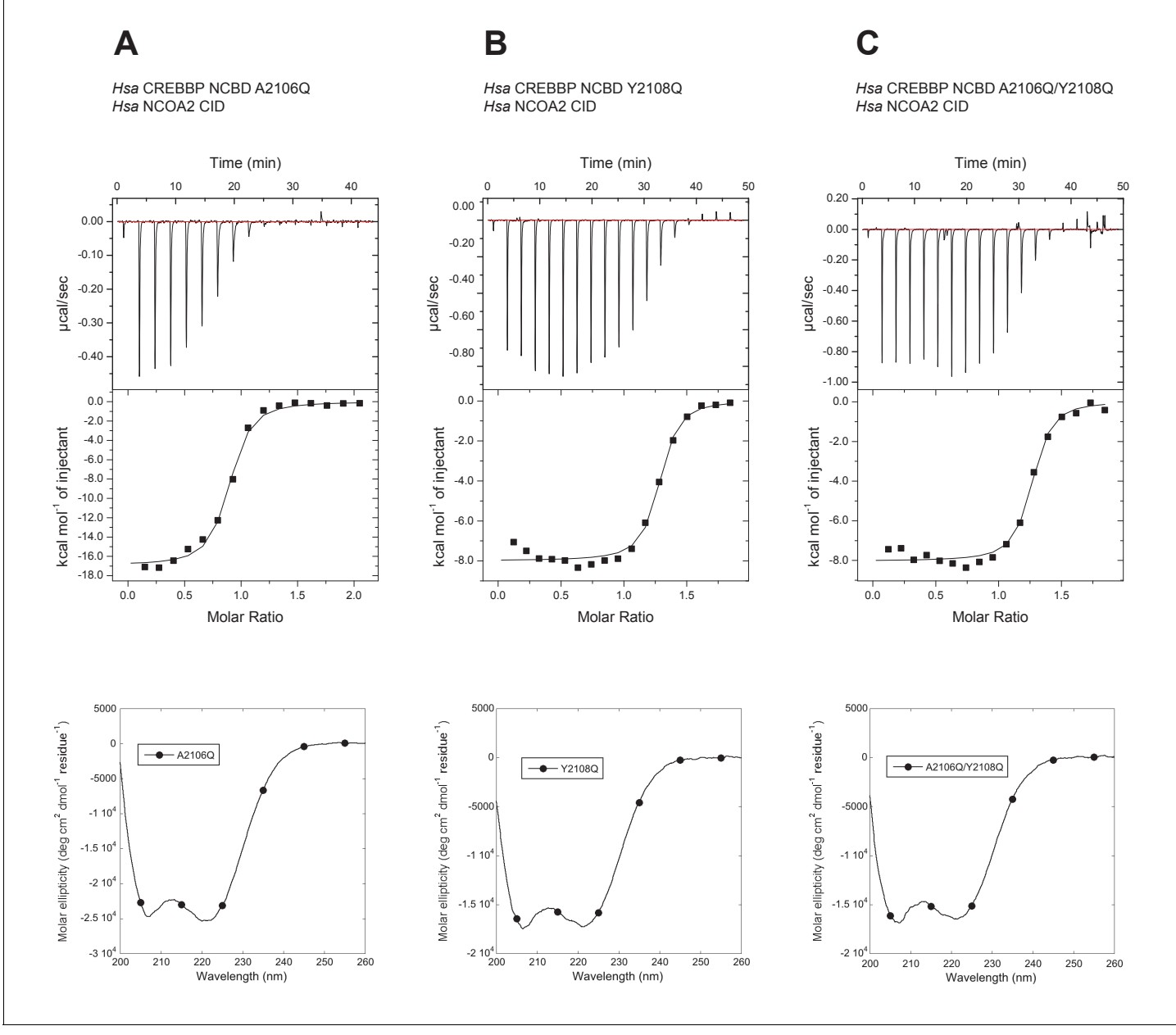

**Figure 12.** Isothermal titration calorimeter experiments between human NCOA2 CID and 'reverse mutants' in human CREBBP NCBD. (A) A2106Q, (B) Y2108Q and (C) A2106Q/Y2108Q. Below are CD spectra of the respective NCBD variant.

also non-conserved residues in the CID/NCBD interface, for example Phe1055 in NCOA1 CID, which is Ala in NCOA2 CID, Leu in NCOA3 CID and Leu in the ancestor. Moreover, Lys1059 in NCOA1 CID forms part of the interface in the complex (PDB code 2C52), whereas the corresponding position in NCOA3 CID is a solvent exposed Thr (the ancestral residue) (PDB code 1KBH). In NCOA2 CID this residue is a Phe, which is likely part of the hydrophobic core of the NCBD/NCOA2 CID complex. A third non-conserved interacting residue is position 1053 in Cα1. Here, the ancestral residue is Asp, while in the human domains it is Val in NCOA1 CID, Tyr in NCOA2 CID and His in NCOA3 CID. These are all examples of substitutions, which could be expected to have a large impact on the protein-protein interaction and also on the structure of the complex. Nevertheless, these mutations in the CID domain have minor impact on the affinity for NCBD, illustrating the high tolerance to mutations in IDPs such as the CID domain, with regard to affinity. The tolerance to mutation is also

reflected in the surprisingly similar affinities of human NCBD for CID domains from sea lamprey, zebra fish and human, respectively (*Table 1*). Based on the two available NMR structures of CID/NCBD complexes (*Figure 4*), one explanation is that the CID domain may adopt distinct conformations in the C-terminal region (corresponding to Cα2 and Cα3 in 1KBH) depending on amino acid sequence; however, the most populated simulated structures (*Figure 9*) are all relatively similar to the 1KBH structure. Some structural features appear important for maintaining the CID/NCBD interaction. We have previously shown that the initial native contacts formed during the binding reaction between human NCOA3 CID (ACTR) and CREBBP NCBD are made between residues in the respective α1 of the two domains (*Iešmantavičius et al., 2014*; *Dogan et al., 2013*). Both Cα1 and Nα1 contain leucine-rich motifs, which are very common in protein-protein interactions (*Kobe and Deisenhofer, 1994*) and which are conserved in the CID/NCBD interaction (*Figure 2*). In contrast, in extant fruit fly (*Drosophila melanogaster*), a proteostome, the NCBD domain is preserved but, similarly to D/P NCBD it has a low affinity for the human CID domains (*Table 1*). The leucine-rich motif in Cα1 is mutated in fruit fly NCBD (*Figure 2*) and the C-terminal region is more similar to D/P NCBD. Thus, it is likely the combination of several amino acid substitutions, affecting both direct interactions and structural heterogeneity, that made the 1R/2R NCBD a high-affinity CID binder.

On a functional level there is much more to the interaction than affinity, for example cellular concentrations, solubility, spatial organisation of the proteins and competing ligands. NCBD is particularly interesting in this respect because it has several physiological ligands including the PNT domain from the transcription factor ETS-2 (*Jayaraman et al., 1999*), IRF-3 (*Lin et al., 2001*), and p53TAD. Any mutation that increases the affinity toward the CID domain must maintain the affinity for the other ligands. Intriguingly, in the complex between CREBBP NCBD and IRF-3 (*Qin et al., 2005*), NCBD adopts a completely different conformation as compared to the conformation with CID domains and p53TAD (*Lee et al., 2010*). Furthermore, NMR and folding experiments suggest that NCBD can adopt two different conformations in absence of ligand, one of which is the CID-bound conformation and the other possibly the IRF-3 bound conformation, although this has not been experimentally confirmed (*Kjaergaard et al., 2010a*, *2013*; *Dogan et al., 2016*). Such functional conformational heterogeneity in NCBD would obviously put extra constraints on the evolutionary process. With these observations in mind, we note that the affinity of the extant human PNT domain of ETS-2 is similar for ancient and extant NCBD domains (~1 μM, *Table 1*). Although we have not resurrected the ancient versions of the PNT domain, these results suggest that the PNT/NCBD interaction was present before, and preserved during the evolution of the CID/NCBD interaction. Consistently, the PNT domain is present in protostomes (e.g. *D. melanogaster*), while the CID domain is not. We observe a similar pattern for the affinity between extant human p53TAD and ancient and extant human NCBD variants. The limit to a further increase in affinity between CID and NCBD in the animal lineage leading to the species that experienced the 1R whole genome duplication could well be due to constraints imposed by the ancient versions of PNT, IRF-3 and p53TAD. On a residue level, it is clear that NCBD employs distinct residues for the interaction with IRF-3, which allows this interaction (and the alternative conformation of NCBD) to coexist with those of CID and p53TAD (*Figure 11*). For CID and p53TAD, similar residues have been used throughout evolution for interactions, albeit with different relative influences.

## Materials and methods

### Ancestral sequence resurrection

CREBBP and p300 nucleotide sequences were downloaded from the Gene tree ENSGT00730000110623, automatically generated by Ensembl (*Flicek et al., 2013*) (www.ensembl.org). This collection was complemented with hits from tblastn (RRID:SCR:011822) searches from preEnsembl (RRID:SCR_006766) (pre.ensembl.org), EnsemblMetazoa (RRID:SCR_000800) (metazoa.ensembl.org), NCBI (www.ncbi.nlm.nih.gov) (Metazoa specific searches), Skatebase (RRID:SCR_005302) (*Wang et al., 2012*) (skatebase.org), JGI Genome Portal (RRID:SCR_002383) (*Grigoriev et al., 2012*) (genome.jgi.doe.gov), MOSAS amphixus (http://mosas.sysu.edu.cn/genome/), EchinoBase (RRID:SCR_007441) (http://spbase.org) (*Cameron et al., 2009*), Japanese lamprey genome project (http://jlampreygenome.imcb.a-star.edu.sg/) (*Mehta et al., 2013*) and OrcAE (http://bioinformatics.psb.ugent.be/orcae/) (*Sterck et al., 2012*). The same was done for

NCOA1, 2 and 3 (Gene tree ENSGT00530000063109). A number of cycles with alignments and maximum likelihood tree analysis were performed to detect erroneous sequences or alignments. The erroneous sequences were manually edited where possible, by for instance removing or adding exons. If manual editing still did not result in a good alignment for that individual gene, the sequence was removed from the analysis. Sequences shorter than 500 amino acids were excluded from the CREBBP analysis (the whole length of the protein is roughly 2400 residues), while sequences shorter than 300 amino acids were excluded from the analysis for the roughly 1400 residues long NCOA proteins. Sequences lacking a complete NCBD or CID domain were also removed. The final sequences were aligned at Guidance (*Penn et al., 2010*) (guidance.tau.ac.il) using MAFFT (RRID: SCR_011811) and ClustalW (RRID:SCR_002909) with the codon model, and Muscle with amino acid model. The best alignment according to guidance alignment score was MAFFT and this alignment was therefore used for the continued analysis. Less reliable residues according to guidance (confidence score below 0.5) were masked as X. Masking residues is preferable to removing whole columns since more good data will be retained in the analysis (*Privman et al., 2012*). All columns without a guidance confidence score, as well as regions with few aligned sequences were also removed with Gap Strip/Squeeze v 2.1.0, which kept columns with less than 95% gap. The amino acid and nucleotide models that best describe the resulting alignment according to the Bayesian information criterion (BIC) (*Schwarz, 1978*), calculated with Mega 5.2.2 (*Tamura et al., 2011*), were JTT+ G+ I and GTR+ G+ I, respectively for CREBBP. For NCOA, the best amino acid model was JTT +G. Using the alignment, maximum likelihood (ML) trees were calculated both for the best nucleotide and amino acid model with PhyML3.0 (*Guindon et al., 2010*) with SPR and NNI tree improvement and with SH-aLRT branch support, which has been shown to perform better than traditional bootstrap. The cnidarian species were chosen as outgroups for the CREBBP tree and non-chordate deuterostomes as outgroups for the NCOA tree. The branches in the tree generated by amino acid alignment follow what has previously been suggested for the evolution of species (*Letunic and Bork, 2011*) and whole genome duplications, while a few branches in the nucleotide tree diverge from the species tree. Our conclusion is therefore that the amino acid model and tree best describe the actual evolution and further analyses were performed using this tree. For ancestral sequence reconstruction, it is very important to have the correct alignment and therefore we cut out the NCBD and CID domains, which should be resurrected, as well as 30 adjacent amino acid residues and realigned the respective set of sequences with MAFFT, Muscle, ClustalW and PRANK with the codon model at Guidance. Sequences that did not have a confidence score above 0.5, for example the urochordata (tunicates) were removed from the CREBBP tree. The final alignments contained 181 CREBBP protein sequences and 184 NCOA protein sequences. The highest guidance alignment score for both the NCBD and CID domains, respectively, were obtained with Muscle. However, for NCBD we obtained an even better alignment score by manually editing the alignment (alignment score of 0.97 versus 0.96 and lowest column score 0.85 versus 0.76). The manually edited NCBD alignment and the CID alignment (*Figure 2—figure supplements 1–2*) were inserted into the respective original full length protein alignments and together with the previously obtained ML trees were used to resurrect the ancestral sequences at Mega 5.2.2, which uses an ML method that correctly deals with indels (*Hall, 2011*).

## Expression and purification of protein domains

The cDNA corresponding to the most likely sequences at each evolutionary node (for both NCBD and CID) were purchased and subcloned into the pRSET vector used previously for expression of both NCBD and CID variants (*Dogan et al., 2012*). All NCBD and CID variants were expressed and purified as previously described using nickel affinity and reversed phase chromatography (*Dogan et al., 2012*). The selected nodes were (*i*) the fish/tetrapod (CREBBP NCBD and NCOA3 CID), (*ii*) 2R, (*iii*) 1R and (*iv*) deuterostome/proteostome (D/P), respectively. When the probability of the resurrected amino acid was lower than 0.7 (*Figure 2—source data 1* and *2*) a construct with the second, third etc. most likely amino acid was produced to investigate possible effects of a different amino acid. In all cases in the present work, there was no significant effect of alternative residues (*Table 1*, *Figure 6*). The numbering of residues for CID domains is according to the splice variant NCOA3-002 (ENST00000371997) and the numbering of residues for NCBD domains according to human CREBBP. The PNT domain of ETS-2 and p53TAD were expressed and purified as previously described (*Dogan et al., 2015*).

## Binding experiments using ITC

All ITC experiments were performed in 20 mM sodium phosphate, pH 7.4, 150 mM NaCl. Protein concentrations were measured either using the absorbance at 280 nm and calculated extinction coefficients or (if the variant lacked Trp and Tyr residues) with a direct detect IR spectrometer. The ITC measurements were performed at 25°C on an iTC200 (Malvern instruments) according to the instructions of the manufacturer. For each experiment, the respective NCBD and CID variant was dialysed against the same buffer to minimize artifacts due to buffer mismatch. The baseline of the experimental data was adjusted to get the lowest possible chi value in the curve fitting. For all high-affinity interactions ($K_d$<1 µM), good baselines were obtained and the binding stoichiometry was generally around 1. For the low-affinity interactions ($K_d$ = 1–10 µM), only one baseline was obtained and the stoichiometry thus not well determined. For certain variants, the ITC measurements were repeated twice with the same sample (i.e. a technical replication) and in two cases the experiments were repeated with new samples (biological repetition). Both the biological and technical repetitions yielded highly similar results. ITC experiments are very sensitive to factors such as buffer mismatch and experiments with poor baselines jeopardizing curve fitting were excluded from the study. In the cases where a certain amino acid position was reconstructed with low probability (<0.7), the most likely variants were subjected to ITC experiments to rule out that the uncertainty affected the conclusions. For example, seven different variants of D/P NCBD were made and their affinities reported in *Table 1*. Three additional variants of 1R CID were also tested with D/P NCBD with virtually identical $K_d$ values. In the case of D/P NCBD/1R CID, the average value of all 10 reported $K_d$ values in *Table 1* (5.1 ± 1.6 µM) were used to calculate the relative affinity shown in *Figure 5B*. Importantly, all variants tell the same story, namely that the affinity between 1R/2R NCBD and 1R CID (and 2R CID and later variants) is significantly higher than that between the ancestral D/P NCBD variants and 1R CID.

## Urea and TFE experiments

All experiments were performed in 20 mM sodium phosphate, pH 7.4, 150 mM NaCl. Circular dichroism (CD) experiments were performed on a JASCO-810 spectropolarimeter with a Peltier temperature control system. Far-UV spectra of NCBD and CID variants were recorded from 260 nm to 200 nm at 4°C using 30 µM protein. To assess the global stability of NCBD, urea (0–8 M) was added to protein (30 µM)-buffer solutions containing 1 M trimethylamine *N*-oxide (TMAO) at 10°C and the CD signal at 222 nm was measured by taking the average of 61 individual recordings at each urea concentration. To assess the helix propensity of CID variants 1,1,1-trifluoroethanol (TFE) was added to protein (30 µM)-buffer solutions at 25°C and the CD signal at 222 nm was measured by taking the average of 61 individual recordings at each TFE concentration. Data from the urea and TFE experiments were fitted to the equation for a two-state process (folded state in equilibrium with the denatured state) to obtain the midpoint of the transition and the associated cooperativity factor (*m* value) (*Fersht, 1999*). Regarding the TFE titrations (little difference between variants) and urea titrations (perhaps a small difference), selected experiments were repeated (for example the D/P NCBD variant). The reported standard errors in *Tables 2* and *3* come from the curve fitting but in this type of biophysical experiments, which we perform routinely in the lab, such errors are generally very similar to the error from three independent experiments. *Figure 5D* contains data from two independent experiments for urea-induced D/P NCBD denaturation, showing the high precision in these types of experiment. The largest source of error for the transition midpoint in this type of experiment is the concentration of TFE or urea, which are both determined with high accuracy. The urea concentration is double checked by measuring the refractive index. The largest source of error in the free energy of unfolding $\triangle G_{D\text{-}N}$ derives from the error in the $m_{D\text{-}N}$ value (describing the 'cooperativity' of the transition). The error in the $m_{D\text{-}N}$ value is large because of the small size of the proteins in the present study, resulting in short baselines and low accuracy in the parameters derived from the curve fitting. However, similar to the ITC experiments, the results are clear insofar that there are only small insignificant differences between most variants. The D/P NCBD (and *D. melanogaster* NCBD) have a slightly lower midpoint for urea denaturation, and lamprey NCBD a slightly higher one, but given the error associated with $m_{D\text{-}N}$, and the lack of well-defined baselines (due to the broad transition), we do not stress this apparent difference in $\triangle G_{D\text{-}N}$ as a major finding.

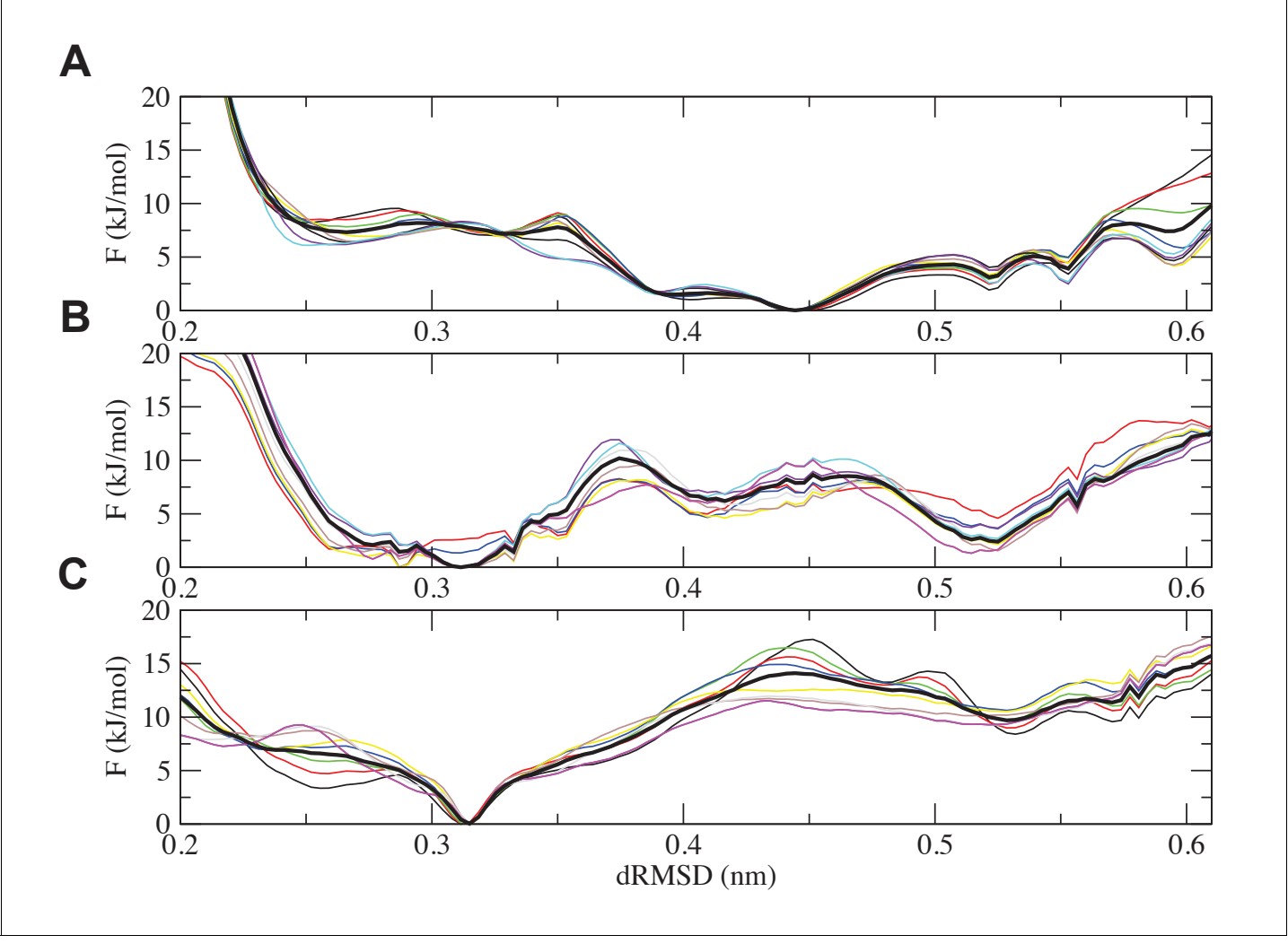

**Figure 13.** Analysis of the convergence of the simulations. Free-energy plots along the dRMSD collective variable for the second half of the simulations are shown for (**A**) the most ancient complex (1R CID and D/P NCBD), (**B**) the 1R/2R complex and (**C**) the extant NCOA3 CID/CREBBP NCBD complex. The solid black curve is the final free energy obtained by averaging over the second half of the simulations. Overall, all the simulations are converged within 3 kJ/mol.

## NMR spectroscopy

The respective CID and NCBD domains were expressed as unlabeled (in rich medium) and $^{15}N$-$^{13}C$ doubly labeled (in minimal medium containing $^{15}NH_4Cl$ and $^{13}C$-D-glucose). Purification was as previously described for unlabeled protein (*Dogan et al., 2012*). After purification, the samples were lyophilized and stored at −20°C. Prior to use the lyophilized samples, they were dissolved in a buffer containing 10 mM sodium phosphate pH 6.8, 150 mM NaCl and dialyzed against the same buffer overnight at 25°C. Protein complexes were formed by titrating saturating concentrations of the unlabeled (CID or NCBD) to labeled fractions (NCBD or CID). The final NMR samples contained 300–500 μM of the labeled protein (bound) to which 0.01% $NaN_3$ and 5% $D_2O$ was added. NMR experiments were acquired on Bruker 600, 700 and 900 MHz spectrometers equipped with triple resonance cryogenic probes at 25°C. For assignment purposes, standard 3D HNCACB and $^{15}N$-resolved NOESY-HSQC (*Cavanagh et al., 2007*) were recorded. All experiments were processed with NMRPipe (*Delaglio et al., 1995*) and analyzed with CcpNmr (*Vranken et al., 2005*). The backbone chemical shifts were used in the simulations as described in the next section.

## Ensembles determination

Structural ensembles of the CID/NCBD complexes were obtained using Metadynamic Metainference (*Bonomi et al., 2016a*, *2016b*). Prior information contained in the Charmm22* force field (*Piana et al., 2011*) with explicit solvent. Chemical shifts (C$\alpha$, C$\beta$, N and H) were included using Metainference in its Gaussian form (*Bonomi et al., 2016a*) using CamShift (*Kohlhoff et al., 2009*; *Camilloni et al., 2012b*) over N = 10 replicas, r. The Metainference energy was calculated as:

$$E = \sum_{r=1}^{N} \left\{ E_{ff} + k_B T \sum_{i=1}^{N_d} \left[ \frac{(f_i(X) - d_i)^2}{2\sigma_{r,i}^2} + 0.5 log 2\pi\sigma_{r,i} + 0.5 log\sigma_{r,i} \right] \right\}$$

where the first term is the sum the over the replicas and the second term is the sum over the experimental data. $E_{ff}$ is the energy of the force-field, $k_B T$ is the Boltzmann constant times the temperature, $f(X)$ is the calculated chemical shift averaged over the replicas, $d$ is the reference experimental value, $\sigma$ is the error estimated on-the-fly that includes the standard error of the mean resulting from the averaging over a finite number of replicas as well as the error estimate for random, systematic and the intrinsic error of the forward model (i.e. CamShift). All simulations were run in GROMACS (*Pronk et al., 2013*) using PLUMED 2 (*Tribello et al., 2014*). Van der Waals and Coulomb interactions were implemented with a cutoff at 0.9 nm, and long-range electrostatic effects were treated with the particle mesh Ewald method on a grid with a mesh of 0.1 nm. All simulations were carried out in the canonical ensemble at constant volume and by thermosetting the system using a stochastic velocity rescaling (*Bussi et al., 2007*). The starting conformations were taken from the available NMR structures (PDB code 1KBH and 2C52) and mutated accordingly using Scwrl4 (*Krivov et al., 2009*). The structures were solvated with 5000 water molecules and ions were added to neutralize the total charge. Two preliminary 100 ns long simulations were run for each structure to equilibrate the system. Metadynamic Metainference simulations were performed using 10 replicas. The sampling of each replica was enhanced by Parallel Bias Metadynamics (*Pfaendtner and Bonomi, 2015*) along five collective variables (CVs) namely, the helix content of CID, the helix content of NCBD, the radius of gyration of the complex, the dRMSD from 1KBH calculated using only the C$\alpha$ carbons and the AlphaBeta collective variable defined as one half of the sum over all residues of one plus the cosine of $\chi_1$ angles for all hydrophobic residues, except alanine. Gaussians deposition was performed with an initial rate of 0.2 kJ/mol/ps, where the $\sigma$ values were set to 0.2, 0.1, 0.01, 0.05, and 0.5 for the five CVs, respectively. In order to keep under control the convergence of the simulations, we rescaled the height of the Gaussians using the well-tempered scheme with a bias-factor of 16 (*Barducci et al., 2008*). Furthermore, in order to limit the extent of accessible space along each collective variable and correctly treat the problem of the borders, intervals were set to 8–28, 18–38, 1.2–1.6, 0–0.6 and 0–24 for the five CVs, respectively (*Baftizadeh et al., 2012*). We set the bias as constant outside a defined interval for each CV. Each replica has been evolved for 150 ns. The sampling of the 10 replicas was combined using a simple reweighting scheme based on the final bias B where the weight of a conformation is given by $w = exp(+B(X)/k_B T)$, consistently with the quasi static behavior at convergence of well-tempered metadynamics. An analysis of the convergence of the simulations is shown in *Figure 13*.

## Acknowledgements

This work was supported by the Swedish Research Council.

## Additional information

### Funding

| Funder | Author |
| --- | --- |
| Vetenskapsrådet | Per Jemth |

The funders had no role in study design, data collection and interpretation, or the decision to submit the work for publication.

## Author contributions

GH, Conceptualization, Formal analysis, Supervision, Investigation, Visualization, Methodology, Writing—original draft, Writing—review and editing; EÅ, Data curation, Formal analysis, Investigation, Visualization, Methodology; CC, Data curation, Formal analysis, Investigation, Visualization, Methodology, Writing—original draft, Writing—review and editing; GNS, Formal analysis, Investigation; EA, Investigation, Methodology; JD, Formal analysis; CNC, Formal analysis, Investigation, Methodology, Writing—review and editing; MV, Resources, Formal analysis, Supervision, Investigation, Writing—original draft; PJ, Conceptualization, Resources, Formal analysis, Supervision, Funding acquisition, Investigation, Methodology, Writing—original draft, Project administration, Writing—review and editing

## Author ORCIDs

Greta Hultqvist, http://orcid.org/0000-0002-4136-6792
Celestine N Chi, http://orcid.org/0000-0003-4154-2378
Michele Vendruscolo, http://orcid.org/0000-0002-3616-1610
Per Jemth, http://orcid.org/0000-0003-1516-7228

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
