## [Decision Letter]

Thank you for submitting your article "Emergence and evolution of an interaction between intrinsically disordered proteins" for consideration by *eLife*. Your article has been reviewed by two peer reviewers, and the evaluation has been overseen by Jeffrey Kelly as the Reviewing Editor and Aviv Regev as the Senior Editor. The following individuals involved in review of your submission have agreed to reveal their identity: Rohit Pappu (Reviewer #2).

The reviewers have discussed the reviews with one another and the Reviewing Editor has drafted this decision to help you prepare a revised submission.

The authors pose a very interesting and important question regarding the evolution of IDPs that participate in protein-protein interactions. They combine systematic assessments of sequence alignments with ITC measurements and MD simulations to arrive at some interesting insights regarding the differences between the ancient and extant sequences, especially in relation to the impact of the differences on protein-protein interactions. Overall, this is an interesting study

Summary:

This manuscript needs major revision to be seriously considered for *eLife*. Both reviewers agree that the big picture gets lost in the maze of all the details about specific residues and that we think this can potentially be remedied with some clever restructuring. The figures too need a lot of work. Many of the figures are just background figures and can be moved to the supplemental materials section. We find that the main assertions of the paper that hang mostly on the molecular dynamics simulations, could be better validated and are therefore not well supported. The MDS themselves suffer from a major drawback, which is the assumption that the reconstructed complexes have the same structure as extant ones. As a control, it might make sense for the authors to anchor their choice of structures in a suitable alignment of the structures adopted various bound IDPs as a function of sequence distance to establish the rationale for their assumption.

Essential revisions:

1) We urge caution regarding making pronouncements about the evolution of IDPs. I am familiar with the work that's been cited. Some skepticism regarding these pronouncements might be warranted. And indeed the authors are well poised to test the specific validity of the oft-quoted pronouncement regarding IDP evolution. They could set up a comparison between their two IDP sequences and two autonomously foldable counterparts of equivalent molecular weights. In fact, they can perform internal comparisons between the IDRs and well folded domains of NCBD.

2) There is a second refrain that is often made in the field of IDPs that this work is perfectly situated to test and critique. It is often stated, without compelling evidence, that the sequences of IDPs evolve rapidly (cf., the citations to the evolutionary work made in the manuscript). However, it is also argued that the amino acid compositions show greater conservation than the actual sequences. To what extent is this separation between the degree of variation / composition true in the current exercise and what do the comparisons between the variations of composition vs. sequence tell us about the evolution of biases that determine the properties of CID and NCBD?

3) The interpretation of the metadynamics results raises some questions. The data in Figure 12 are used to assert that CID/NCBD complexes "become less dynamical upon evolution". I see a different and potentially more interesting interpretation and would like the authors to consider this option as an alternative – at least as a comparative alternative. It appears, to the naked eye, that the projected free energy surface undergoes a bifuraction to generate bistable states as the sequences become more recent. Indeed, this bifurcation is manifest along the dRMSD axis and not along the Rg axis, which is not unreasonable. The simplest way to approach the bistability analysis is to fit a mixture of Gaussians to the free energy projection along the dRMSD axis and perform a numberical analysis of the quality of the fit using a K-L measure of divergence from a unimodal vs. a bimodal distribution. If the best description is a bimodal one, then it is likely that the ancient sequence sets up the symmetry breaking to generate a bistability that is adopted by the more recent sequences. As an additional point, the statement regarding increased heterogeneity (one should really call it this and not talk about dynamics because the use of the word dynamics without annotation by timescales is unsatisfying) needs to be quantified. There are several groups that have put forth elegant ideas for quantifying conformational heterogeneity. Stultz's group has used an information theoretic approach, Papoian's group has used a contact-based approach anchored in the tenets of spin glass theory, and Pappu's group has used an approach based on comparative assessments of dihedral angle distributions. The latter's PHI parameter seems well suited for making the point that the authors make in the current manuscript.

4) There is a potentially erroneous simplification that bears scrutiny. The authors focus on the evolution of the affinity of CID-NCBD interactions. However, these affinities need to be placed in the broader context i.e., in the context of the affinity of NCBD for other ligands. The authors touch upon this in the Materials and methods section where they note that "any mutation that increases the affinity toward the CID domain must maintain the affinity toward other ligands". I think this logic need closer quantitative scrutiny. The overall sentiment of specificity being determined by the collection of comparative affinities is spot on. However, if the affinity ∆G were to change for one complex, and the affinities for the other complexes remained invariant, then the specificity will change through evolution because the collection of ∆∆G values will be such that a change in the affinity for one complex, while keep all others fixed will change the vector of ∆∆G values, thus increasing the specificity of NCBD toward the CID domain and away from the others. Is this the idea that the authors wish to communicate? If yes, then the onus is on them to provide an evolutionary trace that shows how ∆∆G changes as a function of mutations thus highlighting the impact of mutations on altered specificities without altering a whole host of affinities. On the other hand it is conceivable that the affinities change and do so in concert to ensure a conservation of specificities and this would point to neutral drift.

5) The general comment about taking on lots of contacts that are weak within the ancient complexes that are swapped for fewer, tighter contacts does not appear to be supported by substantive quantitative analysis. This could be remedied in a revised version.

6) The figures in the paper should be improved. We understand that *eLife* doesn't have strict limits on figure number, but I think the authors should carefully edit the figures. Some of them (e.g. Figure 3, Figure 4, Figure 5, Figure 6) aren't readable because of the volume of data shown. These are probably better left for the supplement, and sensible summaries/reductions could be made into main figures (e.g. a sparser tree, etc.). Additionally, figures are cited in a non-linear order in the text and this is not acceptable. Ideally, the reader can move from one figure to the next, with minimal jumping around. Figure 2 is of very poor quality and none of the details are discernible. The color-coding of the positions in the alignment has to be properly explained. These alignments appear to be qualitative in that they lack the necessary quantitative annotation and an accompanying statistical model to provide one with a quantitative sense of the quality and or significance of the alignments. The same critique applies to Figure 3 and Figure 4. We fully recognize the challenges posed by Figure 5 and Figure 5. However, the criticism of a lack of clarity and illegibility stand for these figures as well. The average reader will not see any point to this figure. If it has to be included in the current format, then it has to go to the supplementary material. If it has to be included in the main text, then it needs a lot of work in order to make it legible. Additionally, a detailed caption will be required to walk the reader through the details.

7) The MDS are based on two structures of extant complexes. The authors find that the ancient complex has more plasticity in the MDS. This could be an artifact of the assumption that the ancient complex has an identical structure as the two extant ones. This concern is especially salient because the authors discuss at various points the fact that other complexes formed by these proteins have different structures. Ideally, the ancient complexes would have structures, too.

8) The conclusions based on the MDS are not corroborated with any experimental data. I know the authors tried with the two mutants they made, but the results there were inconclusive. None of the other experiments the authors did really supported the MDS, either. I think they need something (H-D exchange, light scattering, etc.) to validate the conclusions of the MDS.

[Editors' note: further revisions were requested prior to acceptance, as described below.]

Thank you for resubmitting your work entitled "Emergence and evolution of an interaction between intrinsically disordered proteins" for further consideration at *eLife*. Your revised article has been favorably evaluated by Aviv Regev (Senior editor), a Reviewing editor, and two reviewers.

The manuscript has been improved but there are some remaining issues that need to be addressed before acceptance, as outlined below:

*Reviewer #2:*

I have no major concerns. All of the issues that I raised have been addressed satisfactorily with considerable additional work. My hunch about possible bifurcations in the free energy surfaces of recent proteins has been properly invalidated by additional simulations that improve the statistical convergence. The new free energy surfaces reveal an interesting trend that is well supported by the experiments. The NMR data provide a very nice addition and helps make a complete story. I am fully satisfied and was always keen on seeing this published. This should, in my view, happen post haste.

*Reviewer #3:*

1) The authors add an analysis of evolutionary rates in different domains of CREBP and NCOA. They conclude "…for CID and NCBD, functions of the domains… is probably the determining amino acid evolution rate." I don't think the authors have done enough to justify this conclusion (or any conclusion about evolutionary rate). They measure the rate of accumulation of nonsynonymous substitutions and indels. This quantity is related to evolutionary rate but does not take into account the mutation rate, which can differ between loci. Typically, the dN/dS statistic is used to account for mutation rate (for a primer see http://www.nature.com/nrg/journal/v16/n7/full/nrg3950.html). This is a serious technical flaw that must be amended.

2) To deal with problems related to the MD simulations, the authors have included NMR chemical shift information in the simulations. This is a welcome addition, but I couldn't figure out from the text in the Results or the Materials and methods exactly what NMR data was collected. Is there data for all complexes shown in Figures 10/11? Please clarify.

3) A second issue related to the NMR data is that the data are unavailable/not shown in a figure or supplementary table. I would have liked to be able to examine the chemical shift data (e.g. of the ancient vs extant complexes), and also would have appreciated some principled commentary on the quality of the data. I'm not a structural biologist, so I can't provide specific guidance on what exactly might be shown. Thus, I'll leave it to the other reviewer/the editor to decide whether this request is a reasonable one.

4) On the Results section, the authors state "Overall, the main effect observed… is an increase in the strength of the contacts formed by the C-terminal residues." I have two issues with this statement. First, what the authors actually measured was the normalized number of contacts in the simulations. Can the strength of the contacts actually be inferred by the number of contacts in the simulation? If so, it might help the less specialized reader to point this out and cite a reference for them to examine. If not, then the statement should be amended. The second issue I have is that the data seem more complex than the picture painted by this sentence and the next section of the results. For example, position 2105 has a big change but isn't discussed. Also, several positions in the N-terminus also undergo changes in the number of interface contacts. How should the reader think about those changes? Are they unimportant for some principled reason? Or, have the authors just chosen to avoid commenting on them for the sake of brevity?

5) Related to point #4, one of my original concerns was that the conclusions of the MD simulations were not corroborated with experimental data. I appreciate that the authors collected NMR data, which made the MD simulations more accurate. However, they didn't address my original concern which was that the conclusions of the MD simulations aren't well corroborated. The analysis of the mutants, which appeared in the original submission, remains only partially convincing. Additional data to support the conclusions of the MDS (e.g. testing more hypotheses arising from the MDS by mutagenesis, etc.) would make the paper stronger.

---

## [Author Response]

*Summary:*

*This manuscript needs major revision to be seriously considered for eLife. Both reviewers agree that the big picture gets lost in the maze of all the details about specific residues and that we think this can potentially be remedied with some clever restructuring. The figures too need a lot of work. Many of the figures are just background figures and can be moved to the supplemental materials section. We find that the main assertions of the paper that hang mostly on the molecular dynamics simulations, could be better validated and are therefore not well supported. The MDS themselves suffer from a major drawback, which is the assumption that the reconstructed complexes have the same structure as extant ones. As a control, it might make sense for the authors to anchor their choice of structures in a suitable alignment of the structures adopted various bound IDPs as a function of sequence distance to establish the rationale for their assumption.*

While we have performed NMR experiments in the revised version to back the simulations, we want to stress that the main conclusions of the paper regarding for example affinity are based on experiments and not simulation. We have also toned down the conclusions from simulation.

We have chosen to keep the discussion about specific residues in the Discussion section since we think they exemplify well how permissive this particular IDP interaction is to apparently "rough" mutations.

*Essential revisions:*

*1) We urge caution regarding making pronouncements about the evolution of IDPs. I am familiar with the work that's been cited. Some skepticism regarding these pronouncements might be warranted. And indeed the authors are well poised to test the specific validity of the oft-quoted pronouncement regarding IDP evolution. They could set up a comparison between their two IDP sequences and two autonomously foldable counterparts of equivalent molecular weights. In fact, they can perform internal comparisons between the IDRs and well folded domains of NCBD.*

This is a very interesting suggestion and we have now performed such a comparison between disordered and ordered domains within NCOA and CREBBP/p300, respectively. The results are presented in a new Figure 3 and a new paragraph in the Results section. In conclusion, there are differences in the apparent rate of residue substitution between the analyzed domains, but it is not clear if they relate to protein disorder: two out of four tested domains in CBP/p300 have a similar rate as NCBD and one ordered domain in NCOA has a slightly lower rate than CID. For NCOA, difficulties in assigning reliable domain borders precluded a deeper analysis.

We are also more cautious regarding statements about IDP evolution in general in the revised version, since the present study focus on details of a particular interaction rather than a large scale assessment of evolution rates in IDPs.

*2) There is a second refrain that is often made in the field of IDPs that this work is perfectly situated to test and critique. It is often stated, without compelling evidence, that the sequences of IDPs evolve rapidly (cf., the citations to the evolutionary work made in the manuscript). However, it is also argued that the amino acid compositions show greater conservation than the actual sequences. To what extent is this separation between the degree of variation / composition true in the current exercise and what do the comparisons between the variations of composition vs. sequence tell us about the evolution of biases that determine the properties of CID and NCBD?*

Looking at the same well folded and disordered regions within NCOA and CREBBP that we used to address point 1 we observe no clear difference in conservative versus non-conservative substitutions for ordered and IDP domains, respectively. For three domains (CID in NCOA3, PHD and KIX from CREBBP) there might be a bias towards more non-conservative mutations in early evolution as compared to HAT, NCBD, TAZ1 (CREBBP) and Pas-A (NCOA3). However, it is not straightforward to define a conservative mutation since every mutation is context dependent.

Regarding composition, we analyzed it for the ancient and modern variants in the CREBBP and NCOA3 lineages, respectively. With a few exceptions, the compositions are very similar for all domains analyzed.

However, in both these cases the data sets are limited and we would rather not draw any conclusions since the study is not designed to address these questions. We think that the experiments better reflect any changes in properties in CID and NCBD (or lack thereof) during evolution.

*3) The interpretation of the metadynamics results raises some questions. The data in Figure 12 are used to assert that CID/NCBD complexes "become less dynamical upon evolution". I see a different and potentially more interesting interpretation and would like the authors to consider this option as an alternative – at least as a comparative alternative. It appears, to the naked eye, that the projected free energy surface undergoes a bifuraction to generate bistable states as the sequences become more recent. Indeed, this bifurcation is manifest along the dRMSD axis and not along the Rg axis, which is not unreasonable. The simplest way to approach the bistability analysis is to fit a mixture of Gaussians to the free energy projection along the dRMSD axis and perform a numberical analysis of the quality of the fit using a K-L measure of divergence from a unimodal vs. a bimodal distribution. If the best description is a bimodal one, then it is likely that the ancient sequence sets up the symmetry breaking to generate a bistability that is adopted by the more recent sequences. As an additional point, the statement regarding increased heterogeneity (one should really call it this and not talk about dynamics because the use of the word dynamics without annotation by timescales is unsatisfying) needs to be quantified. There are several groups that have put forth elegant ideas for quantifying conformational heterogeneity. Stultz's group has used an information theoretic approach, Papoian's group has used a contact-based approach anchored in the tenets of spin glass theory, and Pappu's group has used an approach based on comparative assessments of dihedral angle distributions. The latter's PHI parameter seems well suited for making the point that the authors make in the current manuscript.*

The reviewers raise a very important point. By repeating the simulations and this time including the information from NMR chemical shifts, we found a very simple answer. First of all the overall differences between the three complexes are reduced as compared to our initial simulation. So, even if the extant complex in the new simulation is less heterogeneous from a structural point of view than the most ancient one, these differences are much smaller than previously observed. Furthermore, and more important, a more detailed analysis of the ensembles resulted in very localized differences corresponding to the amount of helical structure of the N- and C- terminal helices of CID and of the C-terminal helix of NCBD. This result was confirmed by an independent analysis of the chemical shifts based on an analysis using the δ2D software. Given the current better understanding of the structures and dynamics we have redrawn the free energies as a function of the helix content (new Figure 9). This perspective shows a single well defined minimum that is shifted towards slightly larger values in terms of helical fraction.

*4) There is a potentially erroneous simplification that bears scrutiny. The authors focus on the evolution of the affinity of CID-NCBD interactions. However, these affinities need to be placed in the broader context i.e., in the context of the affinity of NCBD for other ligands. The authors touch upon this in the Materials and methods section where they note that "any mutation that increases the affinity toward the CID domain must maintain the affinity toward other ligands". I think this logic need closer quantitative scrutiny. The overall sentiment of specificity being determined by the collection of comparative affinities is spot on. However, if the affinity ∆G were to change for one complex, and the affinities for the other complexes remained invariant, then the specificity will change through evolution because the collection of ∆∆G values will be such that a change in the affinity for one complex, while keep all others fixed will change the vector of ∆∆G values, thus increasing the specificity of NCBD toward the CID domain and away from the others. Is this the idea that the authors wish to communicate? If yes, then the onus is on them to provide an evolutionary trace that shows how ∆∆G changes as a function of mutations thus highlighting the impact of mutations on altered specificities without altering a whole host of affinities. On the other hand it is conceivable that the affinities change and do so in concert to ensure a conservation of specificities and this would point to neutral drift.*

The overall aim of the research programme is indeed to monitor changes in specificity for competing ligands over evolutionary time.

The present manuscript describes our first resurrection of an ancient protein-protein interaction and, since this is a major undertaking, inclusion of more ligands is beyond the scope of the present paper. Therefore, in this manuscript, we cannot provide change in specificity as a function of mutation (or time) for competing ligands. However, we have included a few present day versions of competing protein ligands and we observe similar affinities between for example extant PNT domain and ancient and extant NCBD, respectively.

Thus, what we want to communicate here is the increase in affinity between CID and NCBD while retaining the affinity for the other ligands (based on the extant versions of p53TAD and PNT). We believe this increase in affinity for CID reflects an adaption to a functional affinity given the concentrations of each component, including proteins competing for the interaction. The specificity, i.e., the relative affinity will increase for CID during this period as compared to the other NCBD ligands, but since we have not resurrected PNT and since each historical variant of CID/NCBD involves several mutations we cannot easily quantify ddG as a function of time or mutation but, as for now, prefer a qualitative assessment (Discussion section). It is very clear that key mutations occur between the last common ancestor of deuterostomes (e.g. vertebrates) and protostomes (e.g. insects) and the two whole genome duplications (1R/2R) in the vertebrate lineage, and from then on the affinity has been maintained among vertebrate species.

*5) The general comment about taking on lots of contacts that are weak within the ancient complexes that are swapped for fewer, tighter contacts does not appear to be supported by substantive quantitative analysis. This could be remedied in a revised version.*

In order to address points 5, 7 and 8 we have expressed and purified 13C/15N double labeled CID and NCBD, respectively, for the three complexes characterized by simulation, i.e., two ancient and one extant: 1R CID + D/P NCBD, 1R CID + 1R/2R NCBD and human NCOA3 CID + CREBBP NCBD. The latter one is the same as previously studied by Wright's and Teilum's groups but our constructs are shorter as a result of the phylogenetic analyses. Celestine Chi (new author added to the manuscript) performed NMR experiments to obtain backbone chemical shifts for the three complexes. The chemical shifts were then used to determine new ensembles of structures for the three cases implementing chemical shifts as restraints in the simulations and to cross validate the results using an independent analysis of the chemical shifts based on δ2D. Thus, all the analyses have been repeated showing results qualitatively consistent with those obtained before when employing only MD simulations, but with less marked differences among the three cases. In particular with respect to the contact analysis we have focused it more on the inter-domain regions, because the differences in the intra-domain contacts reflect only the new finding of the slightly increased helix content observed in the extant complex.

*6) The figures in the paper should be improved. We understand that eLife doesn't have strict limits on figure number, but I think the authors should carefully edit the figures. Some of them (e.g. Figure 3, Figure 4, Figure 5, Figure 6) aren't readable because of the volume of data shown. These are probably better left for the supplement, and sensible summaries/reductions could be made into main figures (e.g. a sparser tree, etc.). Additionally, figures are cited in a non-linear order in the text and this is not acceptable. Ideally, the reader can move from one figure to the next, with minimal jumping around. Figure 2 is of very poor quality and none of the details are discernible. The color-coding of the positions in the alignment has to be properly explained. These alignments appear to be qualitative in that they lack the necessary quantitative annotation and an accompanying statistical model to provide one with a quantitative sense of the quality and or significance of the alignments. The same critique applies to Figure 3 and Figure 4. We fully recognize the challenges posed by Figure 5 and Figure 6. However, the criticism of a lack of clarity and illegibility stand for these figures as well. The average reader will not see any point to this figure. If it has to be included in the current format, then it has to go to the supplementary material. If it has to be included in the main text, then it needs a lot of work in order to make it legible. Additionally, a detailed caption will be required to walk the reader through the details.*

We fully agree and these figures were included in the main text only because we were prompted to include them upon submission to *eLife*. In the revised version we have moved all the detailed alignments and tree figures back to the supplementary section and linked them to Figure 2, which contains the alignments of resurrected and a few selected extant sequences as well as the simplified tree. The previous Figure 3–Figure 6 are now called Figure 2—figure supplement 1, Figure 2—figure supplement 2, Figure 2—figure supplement 3 and Figure 2—figure supplement 4, respectively, and we have added captions to Figure 2—figure supplement 3 and Figure 2—figure supplement 4.

Regarding the point that the alignments lack quantitative annotation: We have used state of the art alignment programs but determining the quality of an alignment is notoriously difficult. Currently, the best measure of the quality of an alignment is how good phylogenetic tree it produces. The alignments used throughout this paper produce phylogenetic trees that recapitulate species evolution demonstrating that the alignments are of good quality.

Regarding Figure 2 we apologize for the low quality, which we hope is fixed now (our original figure was of high quality but must have been corrupted along the way). The color coding is now explained in the legend.

Regarding the order of the figures; the figures are now cited consecutively in the text. However, we need to reference back to Figure 5 several times because we think it is valuable to keep the experimental data for binding and stability of the different variants in the same figure.

*7) The MDS are based on two structures of extant complexes. The authors find that the ancient complex has more plasticity in the MDS. This could be an artifact of the assumption that the ancient complex has an identical structure as the two extant ones. This concern is especially salient because the authors discuss at various points the fact that other complexes formed by these proteins have different structures. Ideally, the ancient complexes would have structures, too.*

The simulations have been repeated and now including the chemical shifts as structural restraints in the molecular simulations in order to test better the possibility of observing alternative bound states upon evolution. The resulting ensembles are actually now more similar among each other and more extant-like (to the NCOA3 CID/CREBBP NCBD complex, 1KBH). The differences are essentially related to an increase of the helix content at the N- and C- termini of CID and at the C-terminus of NCBD, in agreement with an independent analysis of the chemical shifts based on δ2D.

*8) The conclusions based on the MDS are not corroborated with any experimental data. I know the authors tried with the two mutants they made, but the results there were inconclusive. None of the other experiments the authors did really supported the MDS, either. I think they need something (H-D exchange, light scattering, etc.) to validate the conclusions of the MDS.*

See replies to point 5 and 7.

[Editors' note: further revisions were requested prior to acceptance, as described below.]

*The manuscript has been improved but there are some remaining issues that need to be addressed before acceptance, as outlined below:*

*Reviewer #3:*

*1) The authors add an analysis of evolutionary rates in different domains of CREBP and NCOA. They conclude "…for CID and NCBD, functions of the domains… is probably the determining amino acid evolution rate." I don't think the authors have done enough to justify this conclusion (or any conclusion about evolutionary rate). They measure the rate of accumulation of nonsynonymous substitutions and indels. This quantity is related to evolutionary rate but does not take into account the mutation rate, which can differ between loci. Typically, the dN/dS statistic is used to account for mutation rate (for a primer see http://www.nature.com/nrg/journal/v16/n7/full/nrg3950.html). This is a serious technical flaw that must be amended.*

This analysis was added as per request from the first round of reviewing (from reviewer #2?). What we intend to shed light on is not evolutionary rate in the sense mutation rate in the DNA (including the dN/dS ratio) but indeed the rate of accumulation of nonsynonymous substitutions and indels, following selection. Thus, reviewer #3 is correct that we have not used the proper nomenclature. We now explain in the text what we look at and we refer to this as 'amino acid substitution rate', which we understand is the correct term when assessing the changes in amino acid sequence following selection.

*2) To deal with problems related to the MD simulations, the authors have included NMR chemical shift information in the simulations. This is a welcome addition, but I couldn't figure out from the text in the Results or the Materials and methods exactly what NMR data was collected. Is there data for all complexes shown in Figures 10/11? Please clarify.*

Yes, we included chemical shifts for the three distinct complexes in *Figures 10*/11. This has been clarified in the text.

*3) A second issue related to the NMR data is that the data are unavailable/not shown in a figure or supplementary table. I would have liked to be able to examine the chemical shift data (e.g. of the ancient vs extant complexes), and also would have appreciated some principled commentary on the quality of the data. I'm not a structural biologist, so I can't provide specific guidance on what exactly might be shown. Thus, I'll leave it to the other reviewer/the editor to decide whether this request is a reasonable one.*

This is indeed a reasonable request and we have included a new supplementary Figure, Figure 9—figure supplement 1 and a supplementary table that are both linked to the existing Figure 9. The table provides all chemical shift data for the three complexes and the figure shows HSQC spectra for the most ancient complex and for the modern human complex, respectively.

*4) On the Results section, the authors state "Overall, the main effect observed… is an increase in the strength of the contacts formed by the C-terminal residues." I have two issues with this statement. First, what the authors actually measured was the normalized number of contacts in the simulations. Can the strength of the contacts actually be inferred by the number of contacts in the simulation? If so, it might help the less specialized reader to point this out and cite a reference for them to examine. If not, then the statement should be amended. The second issue I have is that the data seem more complex than the picture painted by this sentence and the next section of the results. For example, position 2105 has a big change but isn't discussed. Also, several positions in the N-terminus also undergo changes in the number of interface contacts. How should the reader think about those changes? Are they unimportant for some principled reason? Or, have the authors just chosen to avoid commenting on them for the sake of brevity?*

First point; we have rephrased the sentence and removed (the wrong use of) “contact strength”.

Second point; yes we have avoided going into too much detail but agree that position 2105 should be included in the discussion. We have thus extended the discussion to better account for the differences observed at the N-terminus and at positions 2103 and 2105.

*5) Related to point #4, one of my original concerns was that the conclusions of the MD simulations were not corroborated with experimental data. I appreciate that the authors collected NMR data, which made the MD simulations more accurate. However, they didn't address my original concern which was that the conclusions of the MD simulations aren't well corroborated. The analysis of the mutants, which appeared in the original submission, remains only partially convincing. Additional data to support the conclusions of the MDS (e.g. testing more hypotheses arising from the MDS by mutagenesis, etc.) would make the paper stronger.*

We certainly agree that the simulations could be further tested. However, given the quite extensive effort of collecting NMR data we think that further validation is beyond the scope of the current study as detailed below.

The results of the two positions studied by “reverse mutagenesis” (2106 and 2108) likely reflect an inherent property of the system, namely that mutation perturbs not only the local environment but may have allosteric effects and also change the ground states of the protein-protein interaction (i.e., free and bound states). Thus, there is a considerable risk that further testing by site-directed mutagenesis will be inconclusive and not easy to interpret. In other words (and as discussed in the paper), the surprisingly small effect of the Y2108Q mutation in human CREBBP NCBD could be a result of a structural re-arrangement of the CID/NCBD complex. Because of this any further testing of the MD simulations will be a project in itself (involving extensive mutagenesis as well as NMR experiments) and we therefore hope that you will find our current effort sufficient to warrant publication.